# Kirkendall effect-induced uniform stress distribution stabilizes nickel-rich layered oxide cathodes

Ziyao Gao[1,2], Chenglong Zhao [1] ✉, Kai Zhou[1,2], Junru Wu[1,2], Yao Tian[1], Xianming Deng[1,2], Lihan Zhang[1,2], Kui Lin[1,2], Feiyu Kang [1], Lele Peng [1] ✉, Marnix Wagemaker [3] ✉ & Baohua Li [1] ✉

Nickel-rich layered oxide cathodes promise ultrahigh energy density but is plagued by the mechanical failure of the secondary particle upon (de)lithiation. Existing approaches for alleviating the structural degradation could retard pulverization, yet fail to tune the stress distribution and root out the formation of cracks. Herein, we report a unique strategy to uniformize the stress distribution in secondary particle via Kirkendall effect to stabilize the core region during electrochemical cycling. Exotic metal/metalloid oxides (such as $Al_2O_3$ or $SiO_2$) is introduced as the heterogeneous nucleation seeds for the preferential growth of the precursor. The calcination treatment afterwards generates a dopant-rich interior structure with central Kirkendall void, due to the different diffusivity between the exotic element and nickel atom. The resulting cathode material exhibits superior structural and electrochemical reversibility, thus contributing to a high specific energy density (based on cathode) of 660 Wh kg$^{-1}$ after 500 cycles with a retention rate of 86%. This study suggests that uniformizing stress distribution represents a promising pathway to tackle the structural instability facing nickel-rich layered oxide cathodes.

The general demand of electric vehicle with a mileage of at least 500 km and a considerable lifespan has urged the academia and industry community to develop new cathode materials with both high specific energy and long cycling stability. Nickel-rich layered oxide material is emerged as the promising cathode for power battery because of its high specific capacity and relatively low material cost[1,2]. However, it always faces a trade-off dilemma between energy density and cycling performance[3–5]. Increasing the Ni content higher than 95% represents an effective strategy to increase the specific capacity and energy density. While the deep extraction of Li-ions (Li$^+$) in such high-Ni-content layered oxide would induce drastic "lattice breathe" of the layered structure, and thus lead to the microstructural instability as well as the crack formation in the secondary particles. The as-formed

cracks are identified as the major contribution to the deterioration of the overall performance, due to the following aspects: (1) Crack formation hinders charge transport which leads to charge heterogeneity between the outer and inner part of the secondary particles, resulting in a lower discharge depth of the inner part[6–8]. (2) High crack densities can even lead to disconnected islands of active material, directly lowering the battery capacity[3,6]. (3) Infiltration of electrolyte in the cracks triggers additional interface reactions resulting in the formation of rock-salt/spinel phase on the surface[9,10], thus reducing electric and ion conductivity of the grain surface, and thereby increasing the internal polarization[11].

Existing strategies such as surface coating and structural doping are reported to be effective to maintain the mechanical integrity of the

[1]Institute of Materials Research, Tsinghua Shenzhen International Graduate School, Tsinghua University, Shenzhen 518055, China. [2]School of Materials Science and Engineering, Tsinghua University, Beijing 100084, China. [3]Department of Radiation Science and Technology, Delft University of Technology, Mekelweg 15, 2629JB Delft, the Netherlands. ✉e-mail: c.zhao-1@tudelft.nl; penglele@sz.tsinghua.edu.cn; m.wagemaker@tudelft.nl; libh@mail.sz.tsinghua.edu.cn

secondary particles. For instance, surface coating technique refers to modify the surface structure of the secondary particle by coating a functional material/composite layer. Such coating layer, which typically possesses good mechanical strength, could physically stabilize the secondary particles by reducing the stress concentration near the outer surface[6,12–14]. Elemental doping such as using Al, Ti, Zr, Mo...etc., tends to introduce hetero-metal-atom to the crystal lattice of the layered structure near the surface, consequently enhancing the surface robustness of the material during electrochemical cycling[15–17]. Previous studies on cracking mechanism identified that crack formation typically starts in the core region of the secondary particles[18,19], and then propagates throughout the secondary particles to induce the interfacial side reactions. The inherent cause to the cracks is that the core region undergoes higher internal stress during the charge process, and the single secondary particle tends to break into different parts with the highest crack density[3,19]. The structural degradation at the single particle level has also been verified to have profound effects on the electrochemical properties particularly cycling stability[20] and electrochemical kinetics[6]. In this regard, the aforementioned surface coating and structural doping strategies cannot retard the structural degradation in terms of the whole secondary particle, since they only reduce the stress near the outer surface or enhance the surface robustness of the materials. Therefore, it is significantly important to address the crack formation at the core of the polycrystalline secondary particles. Optimizing the stress-strain state and uniformizing the stress distribution within the particle may represent an effective strategy to stabilize the whole particle.

In this work, we develop a unique synthetic strategy involving heterogeneous nucleation and the Kirkendall effect[21] to uniformize the stress distribution in nickel-rich cathode secondary particles. By introducing a heterogeneous oxide seed (such as $Al_2O_3$ and $SiO_2$) for the preferential growth of the secondary particle precursors, the secondary particles (nickel content up to 96%) with interior-rich dopant, refined grains, and central void structure are inherited from the precursor due to the different diffusivity between the heterogeneous metal(loid) atom and Ni after the calcination process. Experimental results show characteristics above could effectively alleviate the stress concentration and maintain the structural integrity of the secondary particles, contributing to an ultrahigh specific energy density of 660 Wh kg$^{-1}$ after 500 cycles with a retention rate of 86%. This work suggests that uniformizing the stress distribution in the secondary particle of nickel-rich layered oxide cathodes provides a viable strategy to enable high energy density and high cycle stability in nickel-rich layered oxide cathodes.

## Results

### Heterogeneous nucleation and Kirkendall effect-induced synthesis

The proposed method to synthesize nickel-rich cathodes involves the heterogeneous nucleation on exotic metal/metalloid (EM) oxide seed in a typical co-precipitation process, followed by the solid-state Kirkendall effect in high-temperature annealing (Fig. 1a). In the specific synthetic procedure, EM oxide microparticles such as alumina ($Al_2O_3$) and silica ($SiO_2$) were introduced to the co-precipitation system and acted as the seed for the heterogeneous nucleation of nickel hydroxide precursors. Subsequently, the resulting precursor was lithiated into the final cathode material using a typical solid-state annealing process. Given the fact that the diffusivity of EM atom is considerably larger than that of Ni at high temperature, the solid-state annealing process would promote the Kirkendall effect of EM atom diffusion to the Ni-rich layered oxide cathode, consequently resulting in a single Kirkendall void at the center of secondary particles and an EM-enriched doping interior structure (Fig. 1a).

To demonstrate the feasibility and advantage of the proposed method, $Al_2O_3$ was selected as a model material since Al is considered a

practical and effective dopant for nickel-rich layered oxide cathode materials[22–24]. $Al_2O_3$ with an average particle size of ~1 μm (Fig. S1) was purchased and directly used without any further purification. Due to the introduction of $Al_2O_3$ microparticles, nickel hydroxide precursors would prefer to heterogeneously nucleate on the surface of $Al_2O_3$, rather than to form the individual $Ni(OH)_2$ and $Al_2O_3$ particles. This can be verified by the bonding energy simulation of $Al_2O_3$ to $Al_2O_3$, $xNi(OH)_2$ to $(1-x)Al_2O_3$, and $Ni(OH)_2$ to $Ni(OH)_2$ (Fig. S2 and Research Note I)[25], which corresponds to the cases of agglomeration of $Al_2O_3$, heterogeneous nucleation and homogeneous nucleation, respectively. The bonding energy of $xNi(OH)_2$ to $(1-x)Al_2O_3$ was significantly higher than those of self-bonding energies of $Al_2O_3$-$Al_2O_3$ and $Ni(OH)_2$-$Ni(OH)_2$. The above results suggested that alumina would not agglomerate into larger particle on one hand, and the $Ni(OH)_2$ precursors have a stronger tendency to grow on the alumina rather than forming the individual $Ni(OH)_2$ particles on the other. After the formation of $Ni(OH)_2$ layer on alumina, $Ni(OH)_2$ grains would spontaneously grow on the large secondary particles, driven by the self-bonding energy of $Ni(OH)_2$-$Ni(OH)_2$. Scanning electron microscopy (SEM) at different growth stages was conducted to elucidate the growth mechanism. The pristine alumina microparticles displayed smooth surface, and as reaction time prolonged, the surface of the alumina became rougher (Fig. S3) and the size of the particle gradually increased. When the co-precipitation process completed, the precursors showed uniform size distribution with an average size of ~7 μm (Fig. S4). Ion thinning was further carried out to slice the precursor particles to examine the inner structure. SEM and energy-dispersive spectrometer (EDS) mapping results (Fig. 1b, c) revealed that the resulting secondary precursor particles possessed core-shell structure with distinct distributions of Al in the core region and Ni in the shell region. SEM images under lower magnification proved the distribution of alumina seed in each precursor particle (Fig. S5). The above discussions could validate the growth tendency.

The following calcination process promoted the diffusion of exotic Al atoms in nickel precursors to form the hk-$LiNi_{0.96}Al_{0.04}O_2$. For fair comparison, the $LiNi_{0.96}Al_{0.04}O_2$ counterpart with same composition was synthesized by annealing the nickel hydroxide precursors obtained by the classical co-precipitation method with stoichiometric ratio of aluminum isopropoxide (denoted as c-$LiNi_{0.96}Al_{0.04}O_2$). Acid-base titration (see Table S2) displayed a comparable amount of residual lithium on the pristine surface of the two samples before cycling. The hk-$LiNi_{0.96}Al_{0.04}O_2$ obtained by the proposed method and the c-$LiNi_{0.96}Al_{0.04}O_2$ counterpart showed nearly identical X-ray diffraction peaks, consistent with highly crystalline $LiNiO_2$ layered structure (Fig. S6a, b). With Al dopants introduced, the lattice parameters of the resulting products displayed a small distortion compared with the standard $LiNiO_2$ crystal. The results of the Rietveld refinement analysis (Table S1) indicated lattice parameters $a = b = 2.872$ Å, $c = 14.190$ Å for the as-obtained hk-$LiNi_{0.96}Al_{0.04}O_2$ and $a = b = 2.875$ Å, $c = 14.189$ Å for c-$LiNi_{0.96}Al_{0.04}O_2$. Noteworthy is that a larger ratio of the lattice parameters $c$ to $a$, observed for hk-$LiNi_{0.96}Al_{0.04}O_2$, are associated with a higher Li$^+$ conductivity[22]. Another observation was that the (003) XRD reflection of hk-$LiNi_{0.96}Al_{0.04}O_2$ was broader as compared to that of c-$LiNi_{0.96}Al_{0.04}O_2$ (Fig. S6c). This is most likely the result of (1) the different Al concentration within particle and (2) a smaller crystallite size of the hk-$LiNi_{0.96}Al_{0.04}O_2$ primary particle.

Both synthetic routes yielded spherical secondary particles with a uniform size of ~7 μm in diameter (Fig. S7a–d). Focused ion-beam (FIB) cutting was performed on several randomly picked secondary particles to gain insight on the internal structure. SEM images of c-$LiNi_{0.96}Al_{0.04}O_2$ particle showed a relatively loose structure and a large quantity of small cavities inside the secondary particle (Fig. 1d), which may evolve into sites of crack formation and propagation[7]. Instead, a denser internal structure with no obvious small cavity but a micro-sized central void was observed in the hk-$LiNi_{0.96}Al_{0.04}O_2$

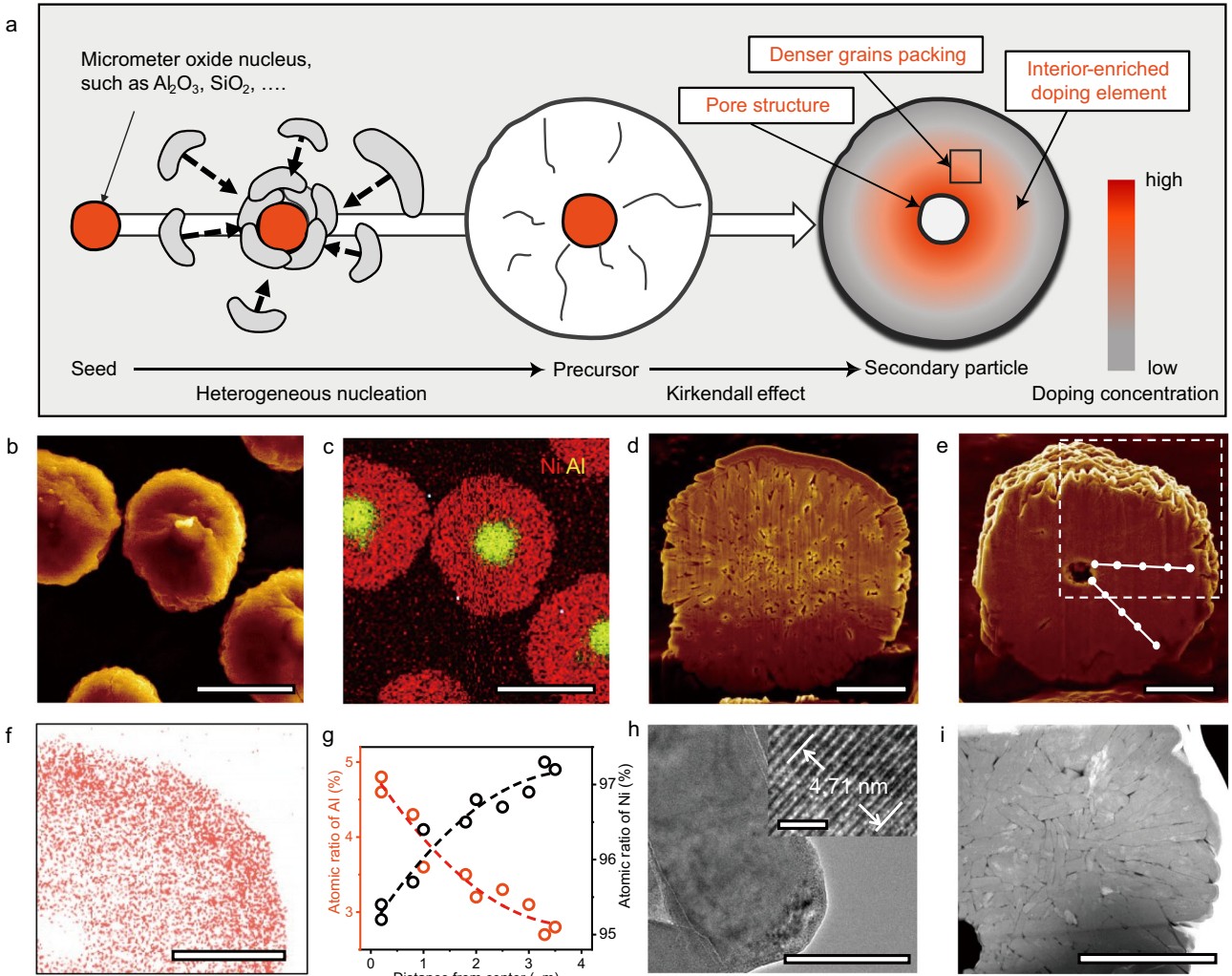

**Fig. 1 | Heterogeneous nucleation and Kirkendall effect-induced synthesis of the hk-LiNi$_{0.96}$Al$_{0.04}$O$_2$ cathode materials. a** Scheme of the proposed heterogeneous nucleation and Kirkendall effect-induced synthesis route. **b** Cross-sectional SEM image showing Ni(OH)$_2$ precursor grown on the Al$_2$O$_3$ nucleus and **c** corresponding EDS mapping photograph during the developed synthesis route. **d, e** Cross-sectional SEM images of the c-LiNi$_{0.96}$Al$_{0.04}$O$_2$ (classical synthesis route) and hk-LiNi$_{0.96}$Al$_{0.04}$O$_2$ (developed synthesis route) secondary particles after calcination. **f–h** Compositional and morphological characteristics of the hk-LiNi$_{0.96}$Al$_{0.04}$O$_2$ secondary particle shown in (**e**), including **f** Cross-sectional Ni mapping image of the hk-LiNi$_{0.96}$Al$_{0.04}$O$_2$ (the dashed square in **e**), **g** the relative content of Al and Ni as a function of the distance from the center of the hk-LiNi$_{0.96}$Al$_{0.04}$O$_2$ particle to the scan point (white spots) on the two scanning routes and **h** primary particle morphology of hk-LiNi$_{0.96}$Al$_{0.04}$O$_2$ in TEM image. The inset displays the HRTEM image of hk-LiNi$_{0.96}$Al$_{0.04}$O$_2$ from [010] axis. **i** HAADF-STEM image of secondary particle. Scale bar, 5 μm (**b, c**); 2 μm (**d–f, i**); 100 nm (**h**) and the scale bar in the inset of **h** is 2 nm.

sample (Fig. 1e)[26]. Ni distribution obtained from EDS element mapping in Fig. 1f confirmed the gradual increase in Ni concentration from the inside to outside of the secondary particle as a result of Kirkendall effect. This is consistent with the results of the EDS point analysis (Fig. 1g), where these two spot scanning routes reflected a similar trend in Al concentration with almost 5% in the core region and just below 3% on the surface of the secondary particle. Such gradient Al doping (Fig. S8a) is due to the different diffusivity of the Al and Ni atoms (Research Note II), i.e., Kirkendall effect[21]. Al doping has been regarded as an effective strategy to reduce the volume effect of lattice during the (de)lithiation, and such modulation is more significantly as the doping content increases (below 6%)[23,24]. Since crack formation is predominantly initiated in the core region, the Al-rich core can be expected to undergo smaller strain and maintain better mechanical integrity during cycling (Fig. S8b).

The proposed method also enables smaller primary particle size of the hk-LiNi$_{0.96}$Al$_{0.04}$O$_2$. As shown in the transmission electron microscopy (TEM) image in Fig. 1h, the primary particle near the core region features a small grain size. The interior structure of the

secondary particle was further demonstrated by High-angle annular dark-field scanning transmission electron microscopy (HAADF-STEM) image (Fig. 1i), where the primary particles near the surface have strikingly larger size with a rod shape (Fig. S7e, f), further verified by the statistic of the grain size as a function of its distance from the particle center in Fig. S7g, j. This could be attributed to the gradient doping of Al inside the hk-LiNi$_{0.96}$Al$_{0.04}$O$_2$ secondary particles. The introduction of excess Al as well as other metal elements (B, Ta, W, Mo, Nb, Sn, and Sb) was found to limit the primary particle size[24]. In addition, intimate contact between grains can be seen in HAADF-STEM image, which was conductive to the transport of ions and electrons on the interface[11,27], thus reducing the mechanical pulverization and homogenizing the charge distribution within secondary particles.

## Simulation of the stress in secondary particles

To verify the hypothesis that the structural features of the resulting cathode material may stabilize the whole secondary particle, finite element simulations were performed to analyze the stress distribution during charging within the particles. To approach the realistic case, the

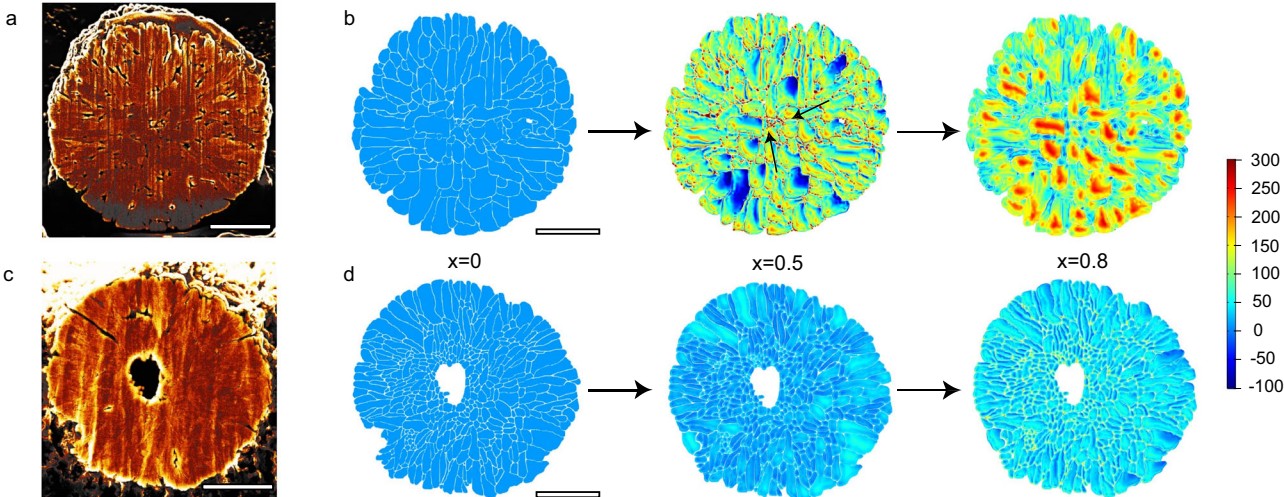

**Fig. 2 | Stress distribution simulation. a, c** Cross-sectional image of **a** c-LiNi$_{0.96}$Al$_{0.04}$O$_2$ and hk-LiNi$_{0.96}$Al$_{0.04}$O$_2$. **b, d** Contour plot of equivalent stress as a function of delithiation depth ($x$ in Li$_{1-x}$Ni$_{0.96}$Al$_{0.04}$O$_2$) within secondary particles upon charging of **b** c-LiNi$_{0.96}$Al$_{0.04}$O$_2$ and **d** hk-LiNi$_{0.96}$Al$_{0.04}$O$_2$. The black arrows show the stress concentration at the core region of c-LiNi$_{0.96}$Al$_{0.04}$O$_2$. Scale bar, 2 μm.

simulation models were constructed by the boundary extraction operation on the SEM image of the two samples (Fig. 2a, c) in AutoCAD. The calculated stress distribution within the c-LiNi$_{0.96}$Al$_{0.04}$O$_2$ and hk-LiNi$_{0.96}$Al$_{0.04}$O$_2$ particles was plotted in Fig. 2b, d as a function of the delithiation depth ($x$ in Li$_{1-x}$Ni$_{0.96}$Al$_{0.04}$O$_2$). It can be seen that both samples showed negligible tensile and compressive strength at pristine ($x = 0$). At a charge depth of $x = 0.5$, the primary particles in the c-LiNi$_{0.96}$Al$_{0.04}$O$_2$ control sample displayed various degrees of tensile and compressive stress, which is mainly distributed between the grains. These black arrows in Fig. 2b ($x = 0.5$) indicate the areas with extremely high tensile strength, which were located predominately in the core region of the secondary particle. It can be considered that cracks are more likely to initiate at the junction of regions with higher tensile stress and compressive stress. Due to the anisotropic shrinkage of the primary particles, the greater tensile strength in the vicinity of the grains would exacerbate the mutual extrusion of the adjacent primary particles, leading to the fast growth and formation of cracks in the center of secondary particles. This result is consistent with previous studies[18,24] With the increase of the charge depth, the tensile strength inside the secondary particle keeps increasing and those highly stressed areas are further transferred to the primary particles in the core region.

In sharp contrast, the tensile and compressive strength distribution of primary particles in the resulting hk-LiNi$_{0.96}$Al$_{0.04}$O$_2$ sample is were much more uniformly distributed (Fig. 2d), either at a relatively low charge depth ($x = 0.5$) or even at a very deep charge depth ($x = 0.8$). The smaller grain near the Kirkendall void structure in hk-LiNi$_{0.96}$Al$_{0.04}$O$_2$ sample could effectively weaken the directionality of the lithium extraction-induced shrinkage, and further uniformize the stress distribution within the secondary particles. The void could offer contraction tolerance and it is believed to a state of compressive stress appears on the surface during the lattice contraction, which is conducive to the maintenance structural integrity of secondary particles. It is worth noting that the average stress in the hk-LiNi$_{0.96}$Al$_{0.04}$O$_2$ sample with a high charge depth of 0.8 is even considerably smaller than that in the control sample with a low charge depth of 0.5. All these simulation results validate the structural features derived from the unique Kirkendall void design and gradient Al doping strategy.

We also experimentally evaluate the mechanical strength of the single secondary particles of the resulting and control cathode materials in the nano-indentation test (Fig. S9a, b). The force-displacement curves plotted in Fig. S9c compare the robustness of the secondary particle for c-LiNi$_{0.96}$Al$_{0.04}$O$_2$ and hk-LiNi$_{0.96}$Al$_{0.04}$O$_2$. With the increase of displacement, the load on the particle would gradually increase until the particle is crushed. It can be seen that the hk-LiNi$_{0.96}$Al$_{0.04}$O$_2$ sample was able to withstand greater load and deformation before crushing, indicating that the resulting cathode with decreased grain size and interior Al-rich doping possesses better mechanical stability than the control sample.

## Structural reversibility and electrochemical reversibility analysis

To explore the structural stability resulting from the unique structure, *operando* XRD was conducted to monitor the changes in lattice parameters, the repeated occurrence of drastic "lattice breathe" at deep delithiation is the origin of crack formation and represents one of the main challenges for layered oxide cathode materials[28,29]. It has been reported that deep charging would induce serious strain at the core region of the particles and consequently the intergranular crack formation in the first cycle due to drastic lattice expansion[4,19,30]. Figure 3a, b presented the evolution of the (003), (101), (104), and (108)/(110) reflections at different delithiation depths ($x$ in Li$_{1-x}$Ni$_{0.96}$Al$_{0.04}$O$_2$) during charge. The H1–H2 transition was characterized by a shift of the (003) reflection to lower diffraction angle, followed by a shift towards a higher angle during the H2–H3 transition[4]. The c-LiNi$_{0.96}$Al$_{0.04}$O$_2$ and hk-LiNi$_{0.96}$Al$_{0.04}$O$_2$ showed a similar evolution of the reflections, with the main difference being that the (003) and (104) reflection of c-LiNi$_{0.96}$Al$_{0.04}$O$_2$ was discontinuous and split into two individual reflections (Fig. 3c). Such two-phase separation implies state-of-charge (SOC) heterogeneity within one single secondary particle[8,20], where one phase underwent drastic lattice contraction/expansion whereas the other not (see Fig. S8c, d). Furthermore, with more Li$^+$ extracted, the gap in lattice parameters $c$ between the two phases continuously widened, which is expected to result in an inhomogeneous strain. Therefore, the abrupt peak shift suggests rapid accumulation of stress in local region, leading to strain concentration during the H2–H3 transition[5,24], which will promote crack formation. Interestingly, the (003) reflection of hk-LiNi$_{0.96}$Al$_{0.04}$O$_2$ underwent a more continuous shift (dashed region) in Fig. 3c, indicating that more uniformly distributed strain and improved reaction homengeneity[31].

Additionally, the shift of the (003) reflection is smaller for hk-LiNi$_{0.96}$Al$_{0.04}$O$_2$ than for c-LiNi$_{0.96}$Al$_{0.04}$O$_2$, reflecting the smaller decrease in lattice parameter $c$ during charging. This was quantified in Fig. 3d, where the lattice parameter $a$ varies similarly, however, the

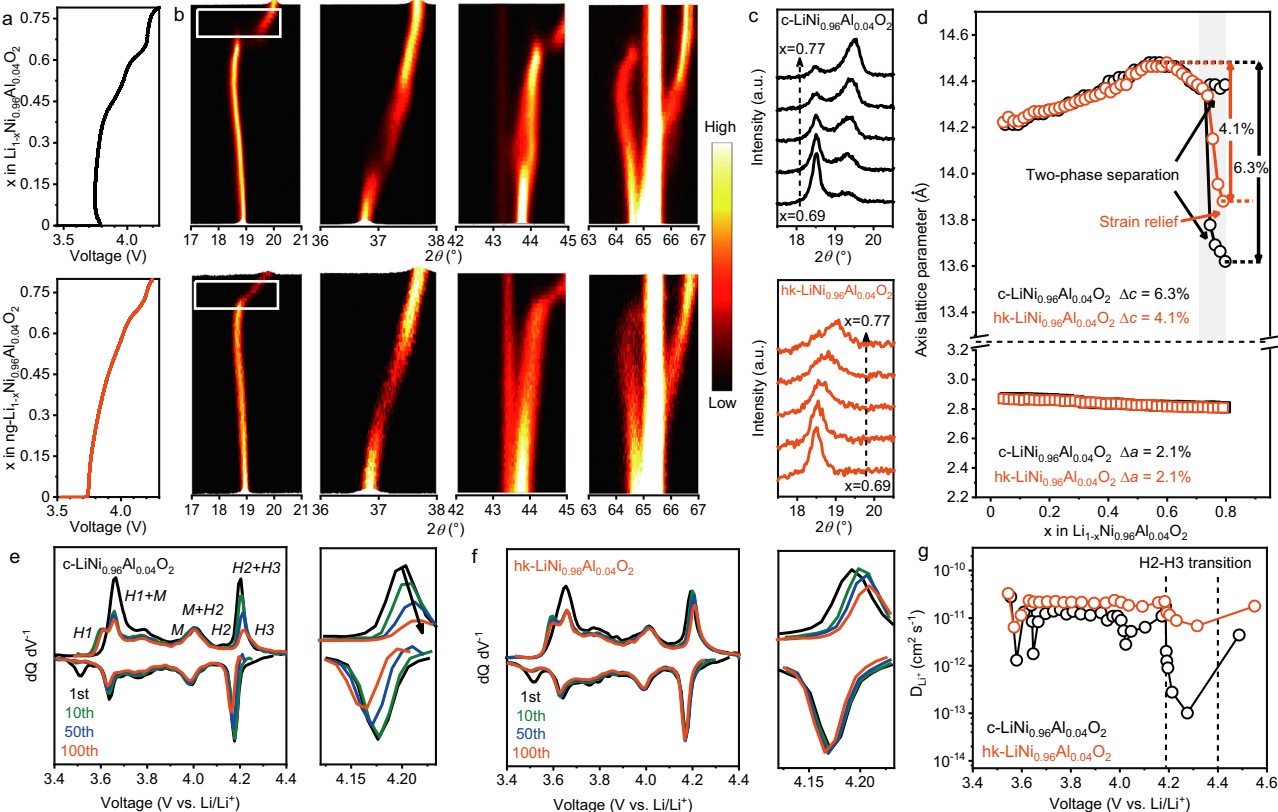

**Fig. 3 | Structural stability and reversibility. a–d** *Operando* XRD of c-LiNi$_{0.96}$Al$_{0.04}$O$_2$ and hk-LiNi$_{0.96}$Al$_{0.04}$O$_2$. **a** voltage curves of c-LiNi$_{0.96}$Al$_{0.04}$O$_2$ (top) and hk-LiNi$_{0.96}$Al$_{0.04}$O$_2$ (bottom) during charging at C/8. **b** Contour plot of the diffraction patterns showing the (003), (101), (104), and (108)/(110) reflections during charging. **c** Magnified images of the dashed squares in the contour plot of (003) reflection, showing the phase separation in a delithiation depth range of x = 0.69 to x = 0.77 of c-LiNi$_{0.96}$Al$_{0.04}$O$_2$ and hk-LiNi$_{0.96}$Al$_{0.04}$O$_2$. **d** Evolution of the a- and c-lattice parameters calculated from the (003) and (101) reflections. The gray area refers to the stage of drastic lattice contraction. **e, f** Differential capacity (dQ dV$^{-1}$) curve of the **e** c-LiNi$_{0.96}$Al$_{0.04}$O$_2$ and **f** hk-LiNi$_{0.96}$Al$_{0.04}$O$_2$ cathodes during cycle between 2.8 and 4.4 V using a constant current of C/3 (60 mA g$^{-1}$) at 35 °C and magnified image within the voltage range corresponding to H2−H3 phase transition. **g** Diffusion coefficient of Li$^+$ in the two cathode materials measured upon charging.

c-lattice parameter decreases to 4.1% for hk-LiNi$_{0.96}$Al$_{0.04}$O$_2$ and 6.3% for c-LiNi$_{0.96}$Al$_{0.04}$O$_2$ at the end of charge (x = 0.8). Such differences in strain could profoundly affect stress state within the secondary particles of layered cathode[6,32,33]. The contour plot of the (003) reflection of hk-LiNi$_{0.96}$Al$_{0.04}$O$_2$ in Fig. S10a, b showed that during the discharge process, hk-LiNi$_{0.96}$Al$_{0.04}$O$_2$ still maintained a continuous single-phase transition, and after a complete cycle, the (003) peak almost unshifted from its original position (Fig. S10c). Besides, high phase transition irreversibility and more interfacial reaction of c-LiNi$_{0.96}$Al$_{0.04}$O$_2$ led to severe Li$^+$/Ni$^{2+}$ mixing up to 4.9% over 500 cycles (Fig. S11a), while that of hk-LiNi$_{0.96}$Al$_{0.04}$O$_2$ showed negligible increase from 2.1% to only 2.3% after 500 cycles in the Rietveld refinement analysis (Fig. S11b), indicating improved structural reversibility[28,34].

The differential capacity (dQ dV$^{-1}$) curves can also be used to investigate the structural reversibility during phase transition for the two samples from the electrochemical perspective. The dQ dV$^{-1}$ curves were shown in Figs. 3e, f. The dQ dV$^{-1}$ curves of both c-LiNi$_{0.96}$Al$_{0.04}$O$_2$ and hk-LiNi$_{0.96}$Al$_{0.04}$O$_2$ samples demonstrated the identical feature within the lower voltage range (3.4-4.1 V) except for the first cycle. This was expected to be the formation of the cathode electrolyte interface (CEI). While at the high voltage region, the dQ dV$^{-1}$ curve of c-LiNi$_{0.96}$Al$_{0.04}$O$_2$ displayed gradually decreased peak intensity in both charging and discharging loops. Notably, the peak positions in the discharging process gradually shifted to the lower potential (Fig. 3e), implying the irreversible structural and consequently elevated cell polarization. This can be also verified by the rapid drop of discharge mid-voltage during 300 cycles of the c-LiNi$_{0.96}$Al$_{0.04}$O$_2$ (Fig. S12). By contrast, the peak shift and intensity decrease of the dQ dV$^{-1}$ peaks of hk-LiNi$_{0.96}$Al$_{0.04}$O$_2$ are almost negligible (Fig. 3f), in line with the steady mid-voltage in Fig. S12. This phenomenon can be attributed to the improved structural stability and reduced crack formation benefitting from the unique structural features achieved by the proposed method.

Galvanostatic Intermittent Titration Technique (GITT) represents a powerful method to provide insights on the Li$^+$ diffusion during the (dis)charging process (Fig. 3g), particularly Li$^+$ diffusion coefficient. Derived from the GITT measurements of both cathode samples, the diffusion coefficients (D$_{Li^+}$) of both samples showed similar order of magnitude (-10$^{-11}$ cm$^2$ s$^{-1}$) in the lower voltage ranges (3.5-4.2 V), in which the layered structure of both sample is well maintained due to continuous and moderate lattice distortion. Nevertheless, the sharp decreased D$_{Li^+}$ of c-LiNi$_{0.96}$Al$_{0.04}$O$_2$ at 4.2 V implies the onset of mechanical degradation and the formation of non-ideal lithium transport phase. Interestingly, in the voltage region where the H2−H3 phase transition occurred, the hk-LiNi$_{0.96}$Al$_{0.04}$O$_2$ exhibited strikingly larger D$_{Li^+}$, which is two orders of magnitude higher than that of c-LiNi$_{0.96}$Al$_{0.04}$O$_2$. This is ascribed to the fact that the higher mechanical integrity reduces interfacial reactions and ensures transport kinetic of lithium-ion between different grains, both of which contribute to a higher D$_{Li^+}$[6,35]. Therefore, combined with the analysis of strain and stress as mentioned, it can be concluded that hk-LiNi$_{0.96}$Al$_{0.04}$O$_2$ demonstrates better electronic and/or ionic contact and transport kinetics of Li$^+$ during the drastic microstructural transition.

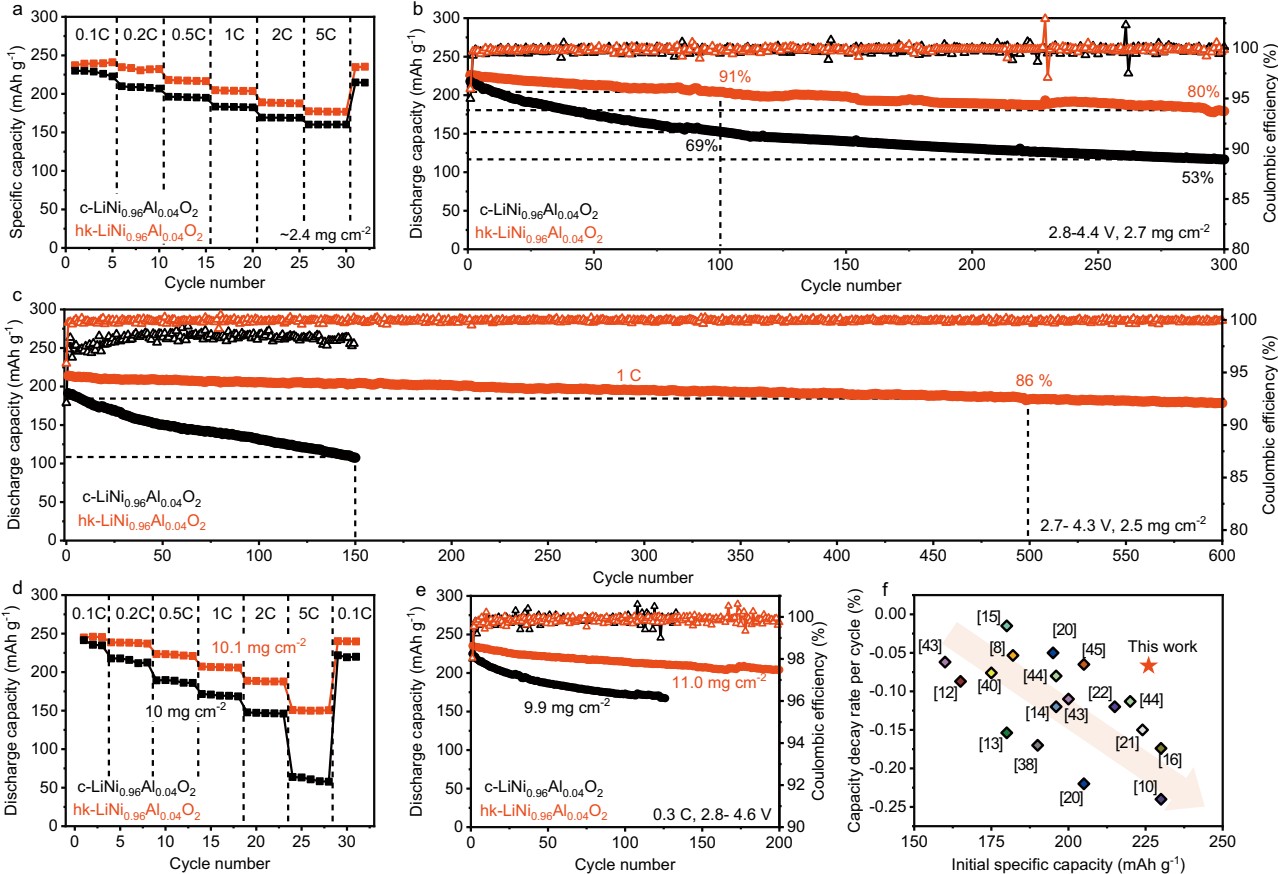

**Fig. 4 | Electrochemical performance. a** Rate performance of c-LiNi$_{0.96}$Al$_{0.04}$O$_2$ and hk-LiNi$_{0.96}$Al$_{0.04}$O$_2$ cathodes in half-cells. **b** Cycling stability of half-cells for hk-LiNi$_{0.96}$Al$_{0.04}$O$_2$ and c-LiNi$_{0.96}$Al$_{0.04}$O$_2$ versus Li$^+$/Li between 2.8 and 4.4 V at a constant current of C/3 at 35 °C. **c** Cycle performance of coin-type full cells with graphite anodes between 2.7 and 4.3 V at 1 C. **d** Rate performance of half-cells under more practical areal loading (about 10 mg cm$^{-2}$) at 0.1 C, 0.2 C, 0.5 C, 1 C, 2 C and

5 C, respectively. **e** Cycling stability of half-cells at high cut-off voltages and more practical areal loading (about 10 mg cm$^{-2}$). **f** Comparison of cycle stability and initial specific capacity between this work (red star) and representative reported nickel-rich cathodes[8,10,12–16,33,34,36,46–50]. Arrows refer to the trade-off dilemma between discharge specific capacity and cycling stability.

These observations can be attributed on the rational Al dopant distribution and better reaction homogeneity resulting from the unique morphology of hk-LiNi$_{0.96}$Al$_{0.04}$O$_2$ secondary particles. The interior-rich Al$^{3+}$ could enhance mechanical strength of the grain boundary and suppress the lattice distortion[22,36], especially in the core region of the secondary particles where the severe phase transformation is considered as a consequence of the higher internal stress[30,37]. Moreover, less lattice distortion reduces the intergranular compatible deformation, and the resulting intimate contact guarantees the maintenance of Li$^+$ transport kinetics between the grains[6,27,35]. reaction heterogeneity within secondary particle has been proposed to accompanied with the stress concentration. More uniform charge heterogeneity and less domain deactivation can also minimize capacity loss, as supported by the observation of continuous phase transitions in *operando* XRD[8]. In addition, the smaller strain of the Al-rich core part is probably due to the central void providing buffer space for the overall contraction and the refined grains enhancing the mechanics property of the secondary particles[31,38,39]. As a result, the crack formation was effectively inhibited at its origin by these structure features.

**Improved electrochemical performance by the uniform stress distribution design**
To identify the advantages of the as-obtained secondary particles with uniform stress distribution, electrochemical performance of hk-LiNi$_{0.96}$Al$_{0.04}$O$_2$ and c-LiNi$_{0.96}$Al$_{0.04}$O$_2$ cathode materials were

evaluated at similar loadings. In Fig. 4a, hk-LiNi$_{0.96}$Al$_{0.04}$O$_2$ showed a higher specific capacity at different C rates from 0.1 C to 5 C, implying that the transport of Li$^+$ was facilitated within the unique structure particularly at a high rate. The charge and discharge curves (Fig. S13a) upon initial activation cycle showed that hk-LiNi$_{0.96}$Al$_{0.04}$O$_2$ delivered a higher discharge capacity of 244 mAh g$^{-1}$ at 0.1 C (1 C = 180 mAh g$^{-1}$) and consequently a higher Coulombic efficiency of 93.6%, compared to 91.5% for c-LiNi$_{0.96}$Al$_{0.04}$O$_2$. As expected, the hk-LiNi$_{0.96}$Al$_{0.04}$O$_2$ sample exhibited a high reversible capacity of 205 mAh g$^{-1}$ after 100 cycles with an excellent capacity retention of 91%, and still delivered 180 mAh g$^{-1}$ after 300 cycles accounting for a high retention rate of 80% (Fig. 4b and Fig. S13c). In sharp contrast, the c-LiNi$_{0.96}$Al$_{0.04}$O$_2$ displayed a severe capacity decay with a retention rate of 69% after 100 cycles and 53% after 300 cycles at C/3 in Fig. S13b.

The cycling stability of coin-type full cells was also investigated (Fig. 4c) with a carefully controlled ratio of negative to positive electrode capacity (N/P ratio). In the case of c-LiNi$_{0.96}$Al$_{0.04}$O$_2$, a severe capacity decay could be observed upon the early cycle, with a capacity retention of 69% over 100 cycles. This is due to the fact that in the full-cell system with limited lithium source, the interfacial reaction caused by cracks mostly contributes to the consumption of available lithium, leading to rapid capacity decay during cycle. In contrast, hk-LiNi$_{0.96}$Al$_{0.04}$O$_2$ exhibits a high initial capacity (208 mAh g$^{-1}$) at 1 C, retaining a retention of 86% over 500 cycles with an average capacity loss of 0.028% per cycle. The inhibition of cracks formation reduces the exposure of fresh surface in electrolyte during repetitive volume

change of cathode upon cycle. The enhanced mechanical stability contributes to less irreversible consumption of Li$^+$ and transition metal dissolution (Ni). As a result, the hk-LiNi$_{0.96}$Al$_{0.04}$O$_2$ exhibits impressive electrochemical stability during long-term cycling.

Furthermore, the electrochemical performance of the two samples was also evaluated under harsher condition of a cut-off voltage up to 4.6 V and practical load (>10 mg cm$^{-2}$) (Fig. 4d, e). At higher cut-off voltage, the deep extraction of lithium ions in the layered structure would induce more drastic lattice contraction, thereby severe volume change, which is a challenge to the mechanical stability of the material. When charged to 4.6 V, the c-LiNi$_{0.96}$Al$_{0.04}$O$_2$ shows a specific capacity of 226 mAh g$^{-1}$ at a current density of 0.3 C. Because of the large volume change at such a high cut-off voltage, the control sample displays a serious capacity decay with a specific capacity of 163 mAh g$^{-1}$ after 120 cycles and a large capacity decay rate of 0.2% per cycle. The inferior structural property of the c-LiNi$_{0.96}$Al$_{0.04}$O$_2$ cathode also results in an inferior rate capability (Fig. 4d). When the current density increasing to 5 C, the capacity quickly drops to 63 mAh g$^{-1}$, probably attributed to the serious cell polarization in the c-LiNi$_{0.96}$Al$_{0.04}$O$_2$ sample. On the contrary, a unique structure designed in resulting hk-LiNi$_{0.96}$Al$_{0.04}$O$_2$ sample could effectively alleviate the crack formation and material polarization at deep charging condition. As a result, the as-developed cathode showed a high specific capacity of 235 mAh g$^{-1}$ at 0.3 C and good cycling stability, still delivering a high specific capacity of 200 mAh g$^{-1}$ after 200 cycles with a small decay rate of 0.07% per cycle (Fig. 4e). In addition, the superior structural property of the resulting hk-LiNi$_{0.96}$Al$_{0.04}$O$_2$ sample also possessed good rate capability, exhibiting 244 mAh g$^{-1}$, 230 mAh g$^{-1}$, 225 mAh g$^{-1}$, 210 mAh g$^{-1}$ and 187 mAh g$^{-1}$ at 0.1 C, 0.2 C, 0.5 C, 1 C, and 2 C, respectively. The resulting cathode could sustain the structural stability when the current density reached 5 C, with a high specific capacity of 150 mAh g$^{-1}$.

Further, the electrochemical performance of the resulting hk-LiNi$_{0.96}$Al$_{0.04}$O$_2$ cathode was carefully compared with other nickel-rich cathodes reported previously in Fig. 4f. Typically, the layered oxide cathode may display decent cycling stability at low delithiation level (low charge depth), which results in a relatively low capacity. As delithiation deepens, higher capacity will be obtained at the expense of deteriorating cycling stability, because the structural instability issue tends to be more prominent, which is a representative capacity-stability dilemma (as indicated in the arrow in Fig. 4f) existing in the nickel-rich cathode. Remarkably, the as-obtained hk-LiNi$_{0.96}$Al$_{0.04}$O$_2$ cathode displayed a high capacity and excellent cycling stability simultaneously owing to its mechanically robust interior structure (stared in Fig. 4f). The upper cut-off voltage is another critical factor for energy density of batteries. As mentioned, the cells using hk-LiNi$_{0.96}$Al$_{0.04}$O$_2$ showed almost no drop in the discharge mid-voltage of (Fig. S12), indicating a lower cell polarization, which is beneficial from less crack formation and electrolyte decomposition. As a result, the high-level energy density of hk-LiNi$_{0.96}$Al$_{0.04}$O$_2$ was stabilized during cycle (684 Wh kg$^{-1}$ for 300 cycles based on active material mass), higher than commercial NCA cathode (487 Wh kg$^{-1}$) in Fig. S14.

## Post-cycling characterization

The internal morphology of the two samples which underwent different stress-strain evolution during (dis)charging was studied. By slicing random selected particles using a focused ion beam (FIB), a large number of intergranular cracks could be observed within the secondary particles of c-LiNi$_{0.96}$Al$_{0.04}$O$_2$ after 100 cycles (Fig. 5a), leading to severe particle pulverization after 300 cycles, as shown in Fig. 5b. In contrast, hk-LiNi$_{0.96}$Al$_{0.04}$O$_2$ showed considerably less crack formation upon cycling as demonstrated by Fig. 5a, b and under lower magnification in Fig. S15. In addition, for hk-LiNi$_{0.96}$Al$_{0.04}$O$_2$, the interior-rich of Al$^{3+}$ could allow the grains in the core region to undergo smaller strain during the (dis)charge as well as enhance the mechanical

strength of grain boundary (Figs. S8 and S9)[24], finally giving the particle a solid and stable core part. Meanwhile, the central void in hk-LiNi$_{0.96}$Al$_{0.04}$O$_2$ provides buffer space for the overall contraction and the refined grains enhance mechanical property of the secondary particles[29]. In line with this, Electrochemical Impedance Spectroscopy (EIS) (Fig. S16) showed that the interfacial charge transfer impedance ($R_{ct}$) of hk-LiNi$_{0.96}$Al$_{0.04}$O$_2$ was significantly smaller than that of c-LiNi$_{0.96}$Al$_{0.04}$O$_2$ at both 100th and 300th cycle (see Table S3). A higher voltage was required for c-LiNi$_{0.96}$Al$_{0.04}$O$_2$ to extract an equivalent number of lithium ions in the preliminary cycles, implying a deteriorated transport kinetics[40]. The above can be interpreted as a result of the better interior integrity of hk-LiNi$_{0.96}$Al$_{0.04}$O$_2$ during cycling, preventing the formation of fresh interfaces and subsequent interfacial reactions.

Moreover, the internal morphology of the c-LiNi$_{0.96}$Al$_{0.04}$O$_2$ and hk-LiNi$_{0.96}$Al$_{0.04}$O$_2$ particles was explored under different delithiation state, where the half-cells were charged to 4.3 V and 4.6 V, respectively. As shown in Fig. 5c, the c-LiNi$_{0.96}$Al$_{0.04}$O$_2$ particle at a charging voltage of 4.3 V suffered from a series of wide intragranular cracks that extended in the radial direction and disconnected part of the secondary particle[6,30]. In contrast, the hk-LiNi$_{0.96}$Al$_{0.04}$O$_2$ particle remained intact at the cut-off voltage of 4.3 V. When further charged to 4.6 V, the cracks in the c-LiNi$_{0.96}$Al$_{0.04}$O$_2$ particles penetrated the secondary particle completely (Fig. 5d), leading to severe contact loss between the secondary particle. Although cracks were also observed in cycled hk-LiNi$_{0.96}$Al$_{0.04}$O$_2$ particles, they were much finer and fewer upon deep delithiation, which can be attributed to the smaller volume change of core part for hk-LiNi$_{0.96}$Al$_{0.04}$O$_2$ particles. Due to the inhibition of excessive interfacial reactions, delithiated hk-LiNi$_{0.96}$Al$_{0.04}$O$_2$ cathode (cut-off at 4.3 V vs Li/Li$^+$) also exhibits higher thermostability (Fig. S17) in the differential scanning calorimetry (DSC) test and *operando* time-resolved (TR)-XRD experiments.

Since the two samples feature different crack-resistance, interface characterizations were employed to compare the resulting interfacial degradation of the two samples[9,41]. X-ray photoelectron spectroscopy (XPS) on cycled positive electrodes shows almost identical peaks shapes of C1s, O1s, and F1s spectra for the two samples (see Fig. S18a–c), indicating similar interface composition of the two samples after 300 cycles as further confirmed by semi-quantitative analysis result (Fig. S18d). The cycled lithium metal anodes were additionally analyzed with time-of-flight secondary-ion mass spectroscopy (TOF-SIMS) to further explore the surface-to-bulk reaction coupling effect[7]. Recent studies found that cracks can provide extra fresh interface and lead to transition metal dissolution (TMD)[9,42]. The dissolution, migration, and deposition of transition metals to anode may damage the solid electrolyte interface (SEI) film, resulting in capacity loss and poor cycle stability of the battery[43,44]. Thus, this phenomenon can be also used to diagnose the structural stability of the different cathode materials. In Fig. 5e, f, signal of Ni species (Ni$^-$ and NiF$_2^-$) was collected on the anode surface for both two samples after 300 cycles. A much stronger signal of Ni$^-$ and NiF$_2^-$ appeared on the anode for c-LiNi$_{0.96}$Al$_{0.04}$O$_2$, reflecting more exposed fresh interfaces of the c-LiNi$_{0.96}$Al$_{0.04}$O$_2$ during long-term cycling. Surprisingly, the signals of Ni species on the lithium anode when coupled with hk-LiNi$_{0.96}$Al$_{0.04}$O$_2$ were barely detective, indicating the superb structural stability during the cycling. Therefore, hk-LiNi$_{0.96}$Al$_{0.04}$O$_2$ sample exhibited considerably better structural reversibility and stability, rendering less crack formation and propagation and consequently an improved cycling stability.

## Discussion
### Extension of the design principle
To demonstrate the wide applicability and universality of the as-developed synthesis approach, SiO$_2$ was also introduced as seeds into the Ni(OH)$_2$ precursors, in this case 2% (mol) of SiO$_2$. SiO$_2$ featuring the

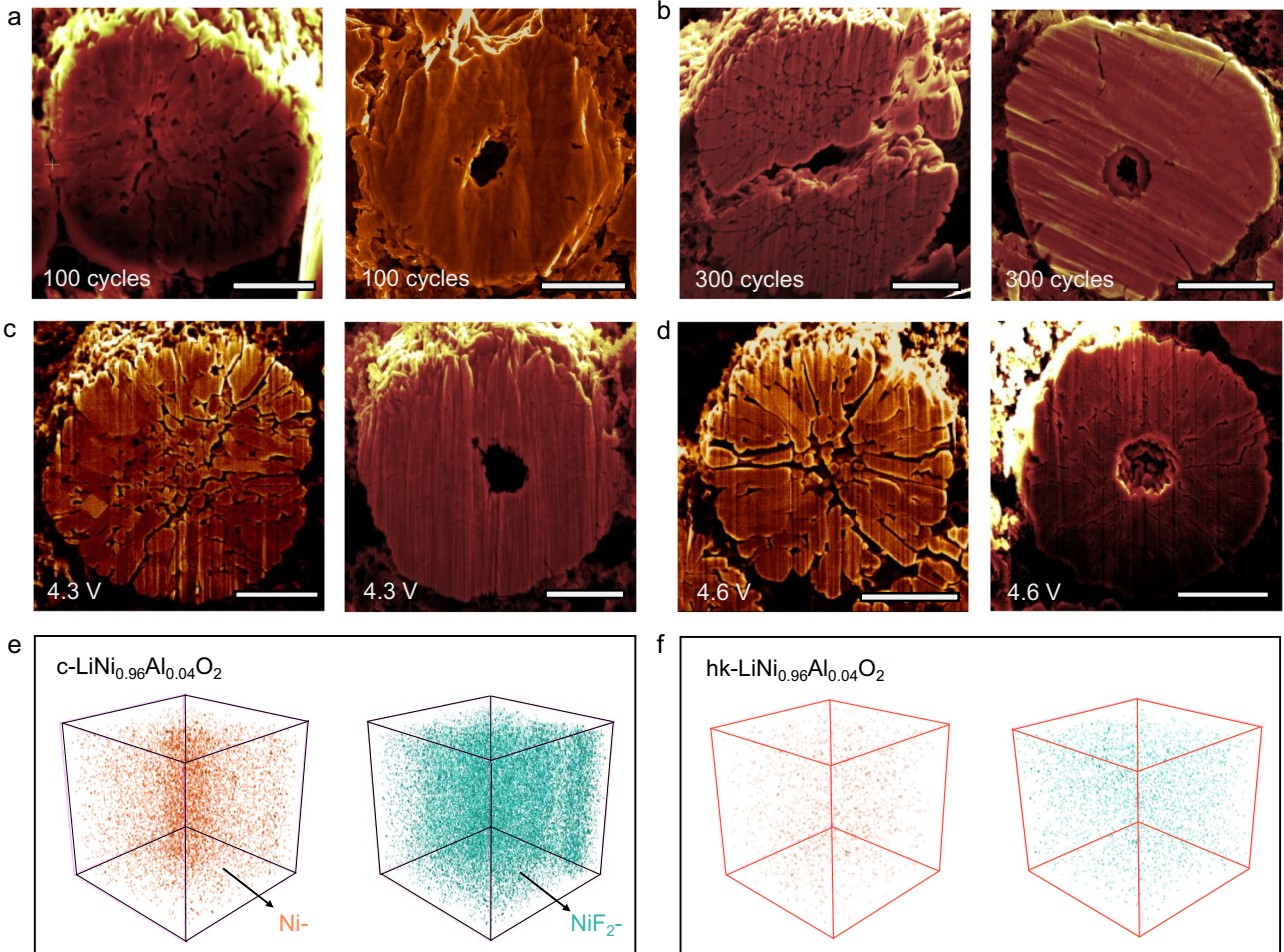

**Fig. 5 | Internal structural stability. a, b** Cross-sectional SEM images of c-LiNi$_{0.96}$Al$_{0.04}$O$_2$ (left) and hk-LiNi$_{0.96}$Al$_{0.04}$O$_2$ (right) after **a** 100 cycles and **b** 300 cycles between 2.8 and 4.4 V using a constant current of C/3 at 35 °C. **c, d** Cross-sectional SEM images of the two particles obtained from half-cells (vs Li/Li⁺) charged to **c** 4.3 V and **d** to 4.6 V at 1st cycle. **e, f** 3D maps of TOF-SIMS data for the cycled lithium metals when using **e** c-LiNi$_{0.96}$Al$_{0.04}$O$_2$ and **f** hk-LiNi$_{0.96}$Al$_{0.04}$O$_2$ as cathode after 300 cycles, showing the distribution of Ni⁻ (left) and NiF$_2$⁻ (right), in sample area 100 × 100 μm with 100 nm depth. The scale bar in (**a–d**) is 2 μm.

advantage of very low cost, simple preparation has been employed as a dopant in previous studies[45] Despite the possible chemical reaction of SiO$_2$ with the alkaline solution, the XRD pattern of the as-obtained hk-LiNi$_{0.98}$Si$_{0.02}$O$_2$ cathode showed a highly crystalline structure, which can be indexed with the $R-3m$ space group (Fig. S19). SEM images of precursors for hk-LiNi$_{0.98}$Si$_{0.02}$O$_2$ were provided in Fig. S20. The internal morphology was explored (Fig. S21a) and element distribution of Si within hk-LiNi$_{0.98}$Si$_{0.02}$O$_2$ particles obtained by calcination, where the point scan analysis results show the signal of Si near the Kirkendall void (spot) formed by high-temperature diffusion in the EDS test (Fig. S21b). Therefore, one can conclude that the as-developed synthesis route can also be appropriate for the introduction of different elements in Ni-rich cathode to improve the material stability, such as for instance titanium, zirconium, magnesium, and manganese.

To explore the applicability of this approach for more complex Ni-based compositions, the Al$_2$O$_3$ seeds were also introduced into the precursors for hk-LiNi$_{0.8}$Co$_{0.15}$Al$_{0.05}$O$_2$ (NCA) in Fig. S22a. A similar structure with precursor for hk-LiNi$_{0.96}$Al$_{0.04}$O$_2$ was obtained as confirmed by the mapping image (Fig. S22b). For comparison, cross-sectional SEM images of the NCA precursor obtained from the classical synthesis under the same conditions are given in Fig. S22c, where the precursors are featured by loose structure. This indicates that the present precursor growth mechanism can be also applied to a wide range of compositions of Ni-rich cathode.

The electrochemical performance comparison was carried out between the resulting and control LiNi$_{0.98}$Si$_{0.02}$O$_2$ as well as LiNi$_{0.80}$Co$_{0.15}$Al$_{0.05}$O$_2$ samples (Fig. S23). As shown in Fig. S23a, b, the hk-LiNi$_{0.98}$Si$_{0.02}$O$_2$ delivered much better capacity at various C rates, indicating its less material polarization (Fig. S23c) and superior rate capability (Fig. S23d). When coupled with the conventional graphite anode with the N/P ratio of 1.16, the full cell based on the hk-LiNi$_{0.80}$Co$_{0.15}$Al$_{0.05}$O$_2$ also exhibits improved cycling stability compared to that based on the control samples (Fig. S23e). And higher specific capacity of hk-LiNi$_{0.80}$Co$_{0.15}$Al$_{0.05}$O$_2$ at 5 C can be seen in Fig. S23f. These results can verify the advantages derived from the as-proposed structural design strategy, and its excellent university to prepare other nickel-rich layered oxide cathode materials.

In summary, a synthesis strategy combining heterogeneous nucleation with the Kirkendall effect is developed to construct the Ni-rich layered oxide cathode for advanced Li-ion batteries. By introducing various EM oxide seeds during the classical co-precipitation, dense and core-shell-structured nickel hydroxide precursors could be obtained. After calcination, Ni-rich layered oxide cathode secondary particle with ultrahigh nickel content (up to 96%), interior-rich dopant, refined grains, and central void structure is formed. This unique morphology and element distribution could effectively mitigate crack formation due to the uniform stress distribution, which addresses the fundamental problem of layered oxide secondary particles prepared

by classical synthesis methods. Experimental results prove improved structural integrity of the hk-LiNi$_{0.96}$Al$_{0.04}$O$_2$ secondary particles, contributing a high specific energy density of 660 Wh kg$^{-1}$ after 500 cycles with a retention rate of 86%. These results provide a new starting point for further development of nickel-rich layered oxide cathode materials, where more in depth studies will help to unlock the full potential of the present approach.

## Methods

### Synthesis
For the developed synthesis route, hydrophilic Al$_2$O$_3$ particles (~1 μm) were dispersed in deionized water by vigorous stirring and supersonic treatment, and subsequently poured into a continuously stirred tank reactor (CSTR) at a stirring speed of 300 rpm before starting the co-precipitation. The CSTR was prefilled with 2 L (0.4 M) of NH$_4$OH (aq) under a replenished N$_2$ atmosphere. Nickel sulfate NiSO$_4$·6H$_2$O was dissolved in deionized water to obtain a 2 M solution and then the solution was pumped into the reactor. Simultaneously, a second feed of solution containing a stoichiometric ratio of 0.4 M NH$_4$OH and 8 M NaOH was injected into the reactor at a constant temperature (50 °C) with the desired PH of ~11.3. The heterogeneous nucleation synthesized Ni$_{0.96}$Al$_{0.04}$(OH)$_2$ precursor materials were collected after a 20 h precipitation reaction. Similarly, other metal oxides seeds can be dispersed in deionized water and added to the reactor, and pumped in to obtain corresponding precursors under the same reaction conditions. For the classical synthesis route, the co-precipitation was conducted under the same reaction conditions as the developed synthesis route, with the difference that no oxide seeds suspend in the base solution. Using a solution-based recipe, a proper amount of aluminum isopropoxide in isopropanol is dissolved, and then added to the Ni(OH)$_2$ precursor, which was continuously stirred at 60 °C until desiccation. The obtained precursors were dried overnight at 120 °C before calcination. The precursors were mixed with LiOH·H$_2$O (Aladdin) in a molar ratio of 1:1.03 and annealed at 500 °C for 5 h, followed by calcination at 700 °C for 10 h under an oxygen flow.

### Characterization
Ion thinning of samples was carried out using Leica EM RES102 at 5 kV for 4 h, and 4.5 kV, 2 h for precursor. The energy-dispersive X-ray spectroscopy (EDS) was conducted on an FEI Quanta 600 FEG at an accelerating voltage of 15 kV. The inductively coupled plasma-mass spectrometry (ICP-MS) was performed on a SPECTRO ARCOS ICP-AES analyzer. The morphologies of the materials were investigated using LEO (Zeiss) 1550 field-emission scanning electron microscopy (SEM) at an accelerating voltage of 5 kV. The cross-sectional samples for SEM characterizations were prepared by an FEI Helios Helios-G4-UC focused ion beam (FIB) operated at 2–30 kV. To protect the sample from beam damage, an 1.5-μm-thick Pt layer was deposited on a particle surface to avoid Ga ion-beam damage in the subsequent lift-out and thinning process. The specimen was thinned to <200 nm. The SEM images were collected in the process of sample thinning. XRD patterns on the precursor and pristine powder materials were acquired through a Rigaku MiniFlex II diffractometer with a Cu Kα radiation (λ = 1.54 Å) X-ray source at a scan rate of 5 °C·min$^{-1}$. Data acquisition was performed from an initial 2θ of 10° to a final 2θ of 80° with a scan rate of 0.20° per minute. The crystal structure refinement was carried out through the Rietveld method, as implemented in the Fullprof software package. And operando time-resolved (TR)-XRD experiments (150 − 400 °C) were conducted on the delithiated cathodes (cut-off at 4.3 V vs Li/Li$^+$) to compare the thermostability of the cathode material. The differential scanning calorimetry (DSC) test was conducted on the delithiated cathodes (cut-off at 4.3 V vs Li/Li$^+$) using differential scanning calorimeter (Mettler DMA1). The nano-indentation experiment was performed by nano-indentation tester (MDTC-EQ-M55-01). XPS testing was carried out using a ULVACPHI 5000 Versa Probe II

instrument equipped with a GCIB sputtering function with a monochromatic Al Kα X-ray source. The sample processed by Helios-G4-UC FIB accessory was transferred to TEM for characterizations when they reach electron transparent thickness. The high-resolution observation, the HAADF-STEM imaging, and EDS mapping were performed using a FEI Tecnai F30 instrument. TOF-SIMS measurements were carried out on a Nano TOF-2 instrument (ULVACPHI, Japan) equipped with a Bi$_3$++ beam (30 kV) cluster primary-ion gun for analysis and an Ar$^+$ beam (3 keV 100 nA) using a sputtering rate of 0.1 nm s$^{-1}$ to obtain the desired depth profile.

### Electrochemical measurement
The electrochemical properties of the samples were measured in 2032 coin-cells which were assembled in the glove box. The slurries were prepared by mixing the active material, acetylene black (conductive additive) and polyvinylidene fluoride (binder) with a molar ratio of 8:1:1, and composite electrodes were obtained by casting the slurries onto the aluminum current collectors. The electrolyte was 1 M LiPF$_6$ with organic solvent ethylene carbonate (EC), diethyl carbonate (DEC), and fluoroethylene carbonate (FEC) in a volume ratio of 1:1:1. The electrochemical performance test was conducted using a NEWARE Battery Test System (CT-4008, Shenzhen, China) in a environmental chamber to ensure constant test conditions. The cycle performance of half-cells was measured in the voltage range of 2.8–4.4 V or 2.8–4.6 V at the current of C/3 (1 C = 180 mA g$^{-1}$), and the rate performance was measured between 2.8 V and 4.4 V at 35 °C. The full cells using graphite negative electrode were tested in the voltage range of 2.7–4.3 V at 1 C. The cells were cycled under C/10 for three cycles as an activation process. The electrochemical impedance spectra (EIS) were recorded with a VMP3 system (BioLogic, France) over the frequency range of 10$^{-3}$–10$^5$ HZ.

### The calculations for the specific energy density
The specific energy (in Wh kg$^{-1}$) is calculated by the following equation:

$$E_{sp} = E/m_{cathode} \qquad (1)$$

where E is the measured value of nominal discharge energy (in Wh) and m$_{cathode}$ is the measured mass (in mg) of the active material.

Further, E is calculated by the following equation:

$$E = \int U \cdot I dt \qquad (2)$$

where U is the measured value of discharge voltage (in V) and I is the current density (in A) of the half-cell (vs. Li/Li$^+$).

### Finite element analysis
The finite element simulations were performed using COMSOL Multiphysics® (Stockholm, Sweden, 1986). The 2D model, which are closer to the realistic case, were constructed by the boundary extraction operation on the SEM image of the two samples in AutoCAD (Autodesk Inc., San Rafael, CA, USA). The diameters of the three models were set to 8 μm. To ensure high accuracy and computational efficiency of multi-physics coupling, finite element simulation based on the diffusion-induced stress theory was performed using COMSOL Multiphysics ® (Stockholm, Sweden, 1986). The extraction/insertion of Li$^+$ is determined by the following diffusion Eq. (3):

$$\frac{\partial c}{\partial t} + \nabla \cdot \boldsymbol{J} = 0 \qquad (3)$$

where $c$ is the molar concentration of lithium, the lithium flux is determined by $\boldsymbol{J} = Dc\nabla\mu/(R_g T)$, $D$, $\mu$, $R_g$ and $T$ are the diffusion coefficient, chemical potential, ideal gas constant, and temperature of

lithium, respectively. The chemical potential can be expressed as:

$$\mu = \mu_0 + R_g T \ln c \tag{4}$$

where $\mu_0$ is the reference chemical potential and $\Omega$ is the partial molar volume of lithium.

Driven by the chemical potential gradient, it is assumed that the insertion/ extraction of $Li^+$ on the surface of the active particle occurs at a constant current of (dis)charge. The corresponding initial conditions and boundary conditions are expressed as follows:

$$c = c_0(t=0) - \boldsymbol{n} \cdot \boldsymbol{J}|_{r=0} = 0, \; -\boldsymbol{n} \cdot \boldsymbol{J}|_{r=a} = \frac{i_n}{F} \text{ (constant current)} \tag{5}$$

where current density on the particle surface $i_n = \frac{nc_m F a}{10800}$, $c_0$ is the initial concentration of lithium in the active particle, $c_m$ is the saturated concentration of the active particle, and n is the current rate upon (dis) charge.

For the electrode particles, which are considered as linear elastic materials, the strain relationship is expressed as follows:

$$\sigma_{ij} = D_{ijkl}\varepsilon_{kl}^e = D_{ijkl}\left(\varepsilon_{kl} - \varepsilon_{kl}^c\right) \tag{6}$$

where $\varepsilon_{ij}^c$ is the diffusion-induced strain generated by the lithiation process, and $\varepsilon_{kl}$ represents the elastic strain, which is determined by the following formula (7).

$$\varepsilon_{kl} = \frac{1}{E}\left[(1+\upsilon)\sigma_{ij} - \upsilon\sigma_{kk}\delta_{ij}\right] \tag{7}$$

where $E$、$\upsilon$、$\sigma_{ij}$和$\delta_{ij}$ were the Young's modulus, Poisson's ratio, stress tensor, and Kronecker symbol of the active material particles, respectively. These mechanical parameters were referenced from previous study.

## Data availability
All the data that support the findings of this study are available from the corresponding author upon reasonable request.

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

## Acknowledgements

This work was supported by the Key-Area Research and Development Program of Guangdong Province (No. 2020B090919003), the National Natural Science Foundation of China (No. 52261160384, 52072208), Shenzhen Basic Research Special Fund (Natural Science Foundation) Key Projects (No. JCYJ20220818101004009), Local Innovative and Research Teams Project of Guangdong Pearl River Talents Program (No. 2017BT01N111), Scientific Research Start-up Funds of Tsinghua SIGS (Grant No. QD2021018C to L.P.) and National Natural Science Foundation of China (Grant No. 20231710015 to L.P.). The authors also acknowledge the Materials and Devices Testing Center at Tsinghua Shenzhen International Graduate School, Shenzhen 518055, China.

## Author contributions

Z.G., C.Z., L.P. and B.L. conceived the idea and designed the experiments. Z.G. and K.Z. conducted synthesis of the material. Z.G., C.Z. and M.W. carried out structural characterization and electrochemical measurements. Y.T. established finite element model. J.W., X.D., L.Z. and K.L. participated in the data analysis and scientific discussion. Z.G., C.Z., L.P., M.W., F.K. and B.L. drafted the manuscript. All authors have given approval to the final version of the paper.

## Competing interests

The authors declare no competing interests.
