## [Peer Review File · Nature Communications]

REVIEWER COMMENTS

Reviewer #1 (Remarks to the Author):

Decision: Reject with minor revision

This work reports the synthesis of $\text{LiNi}_{0.96}\text{Al}_{0.04}\text{O}_2$ through Al_2O_3 oxide seeds that can form heterogeneous nucleation and utilize the Kirkendall effect to generate hollow, Al-gradient doping secondary particles. The modified method can significantly improve the structural stability of $\text{LiNi}_{0.96}\text{Al}_{0.04}\text{O}_2$ by reducing stress buildup at a high de-lithiation stage and alleviating the crack formation of the secondary particles. Compared to the conventional method used to synthesize $\text{LiNi}_{0.96}\text{Al}_{0.04}\text{O}_2$, the above method provides good rate and cycle performance up to 300 cycles in half-cell with Li metal anode. The method can also successfully employ different oxide seeds (SiO_2) to nickel-rich layered oxides. While this manuscript provides some novel findings and shows solid evidence of the structural mechanism, there are a few suggestions that can be considered to improve the manuscript prior to being accepted for publication in Nature Communications. The detailed comments are as follows.

1. It is still unclear how the modified method can give a denser packing density with a smaller particle size. Since the work claims that there is a gradient doping of Al in the secondary particles, why would the dopant affect the particle size to the same degree throughout the particles?
2. In general, according to reported research, the smaller grain size of primary particles will increase the active surface area to react or more interfacial reactions with the electrolyte at a high stage of charge. However, this work has a different discussion that focuses more on the inner bulk of the secondary particles. As the Al dopants from this method locate mostly in the inner bulk, how would this help with the crack formation induced from the surface of the secondary particles? Please explain and include this clarification in the manuscript to avoid any confusion.
3. In Figure 3e, the specific capacity of the c- $\text{LiNi}_{0.96}\text{Al}_{0.04}\text{O}_2$ sample in full cell suddenly drops at around the 35th cycle. Is the performance repeatable? Usually, it should decrease gradually.
4. There are several grammatical errors and typos, please revise the writing more thoroughly. For instance, page 11 has repetitive sentences that can be put in the previous paragraph (Simulation of the stress in secondary particles part). On page 17, Figure S12 needs to be changed to Figure S11 (Improved electrochemical performance by the uniform stress distribution design part). On page 22, it should be secondary particles instead of second particles (conclusion part).
5. One of the main issues for nickel-rich layered oxides is related to thermostability issues at a high stage of charge. It would be good to have a gas release test or DSC data of the samples prepared by the modified method compared to the conventional method.

Reviewer #2 (Remarks to the Author):

The manuscript titled "Kirkendall effect induced uniform stress distribution stabilizes nickel-rich layered oxide cathodes" presents a strategy to uniformize the stress distribution in secondary particle facilitated by Kirkendall effect during the fabrication process to stabilize the high-Ni cathode during electrochemical cycling. As proof of the proposed concept, the authors employed Al₂O₃ as the "seed" to fabricate the yolk-like structured cathode precursor. Subsequently, the calcination process generated a dopant-rich interior structure with a central Kirkendall void. The authors also conducted DFT studies to support this proposition. However, this study lacks the urgency and novelty required for publication in the top journal as Nature Communications. Following some revisions suggested below, it would be more suitable to submit this work to Communications Chemistry for potential publication.

1. The Kirkendall effect has been utilized during the cathode fabrication process to facilitate the generation of primary particles with hollow cores since 2012 (see *Angew. Chem. Int. Ed.* 2012, 51, 239 – 241.). The resulting void structures did enhance the electrochemical stability of the cathode to a certain extent (see Metal segregation in hierarchically structured cathode materials for high-energy lithium batteries *Nat. Energy* 1, 15004 (2016)). The difference of this study: employing a modified protocol, using Al₂O₃ and silica to form a yolk-like structured cathode precursor to enable the dopant-rich interior structure with a central Kirkendall void. The resulting cathode $\text{hk-LiNi}_{0.96}\text{Al}_{0.04}\text{O}_2$ delivers an improved cycling stability but not impressive (especially for the rate performance in Fig. 3a and the cycling performance of Li-ion cells in Fig. 3e) to ensure its publication in top journal as Nature Communications. Unprecedented cycling performance is essential for validating a modified or new strategy/concept.

2. The electrochemical testing and characterization of the resulting cathode $\text{hk-LiNi}_{0.98}\text{Si}_{0.02}\text{O}_2$ are missing, the authors only present the core structure of the precursor rather than present the $\text{hk-LiNi}_{0.98}\text{Si}_{0.02}\text{O}_2$. This compromises the integrity of the work and the range of the developed method to some extent. It is highly recommended to focus on presenting the experimental data rather than speculating on the potential of the current approach.

3. There is an issue with the description of "In the specific synthetic procedure, EM oxide microparticles such as alumina (Al₂O₃) and silica (SiO₂) were introduced to the co-precipitation system and acted as the seed for the heterogeneous nucleation of nickel hydroxide precursors." It is evident that silica is not an exotic metal (EM) oxide. The authors need to correct this and ensure accurate information is conveyed.

4. The caption of Figure 21 is confusing and should be corrected as "Figure S21 a Cross-sectional SEM image and b corresponding element mapping of the Al₂O₃ nucleus introduced into Ni_{0.8}Co_{0.15}(OH)₂ to form the $\text{LiNi}_{0.8}\text{Co}_{0.15}\text{Al}_{0.05}\text{O}_2$ precursor. c, b SEM images and cross-sectional morphology of the precursors for NCA cathode materials obtained by routine co-precipitation method under the same synthesis conditions."

Reviewer #3 (Remarks to the Author):

Remarks to the authors:

This work by Li et al. talks about using Kirkendall effect as a tool to optimize the stress distribution in the $\text{LiNi}_0.96\text{Al}_0.04\text{O}_2$ therefore improve the cycling stability of cathodes. This paper is potentially useful but filled with problems that need to be addressed before being accepted in Nature Communications.

Here are my specific complaints.

1. As being said by the authors, the introduction of Al was reported to decrease the primary particle size, why does the $\text{hk-LiNi}_0.96\text{Al}_0.04\text{O}_2$ has smaller particle size on the surface than $\text{c-LiNi}_0.96\text{Al}_0.04\text{O}_2$, where $\text{c-LiNi}_0.96\text{Al}_0.04\text{O}_2$ is expected to have higher Al concentrations on the surface?
2. From the strain-stress modeling in Figure 2a, I do not see more strains in the center of the particle, this contradicts the explanation in the paper, and in this case, how does model ii and model iii actually help with releasing the strain? Another concern is how primary particle size would have impact on cracking and surface parasitic reactions, also how the strain will actually look like when particle size is decreased, and more grain boundary presents?
3. What is the C-rate being used for the in-situ XRD experiment, is the c-rate slow enough to resolve the peak splitting due to kinetic limitation of phase transition? The OCV for $\text{c-LiNi}_0.96\text{Al}_0.04\text{O}_2$ is slightly higher than $\text{hk-LiNi}_0.96\text{Al}_0.04\text{O}_2$, any explanation for the reason?
4. Middle panel of in-situ XRD in Figure 2 should be 101 peak, while the authors say 104, the author should show the change of 104 as well since it should also split at high SOC. There are many misleading labels of Figures in this manuscript especially in the supporting information, for example, Fig S24, which doesn't exist, Table S2, Fig S12 etc. Many labels should be fixed.
5. Why the failures for half cells and full cells are different, in detail, half cells for $\text{c-LiNi}_0.96\text{Al}_0.04\text{O}_2$ show gradual capacity loss while full cells show a sudden capacity drop after 40 cycles.
6. I believe in industry, people can make good NCA by making NC first and add Al in a solid-state synthesis, can you comment on why Al will form a gradient in your case but not for NCA?
7. In Figure S11, please address where the large ΔV is coming from even in early cycling, $\text{c-LiNi}_0.96\text{Al}_0.04\text{O}_2$ shows larger ΔV than $\text{hk-LiNi}_0.96\text{Al}_0.04\text{O}_2$.

Point-to-point response to Reviewer #1:

***General comments:** This work reports the synthesis of $\text{LiNi}_{0.96}\text{Al}_{0.04}\text{O}_2$ through Al_2O_3 oxide seeds that can form heterogeneous nucleation and utilize the Kirkendall effect to generate hollow, Al-gradient doping secondary particles. The modified method can significantly improve the structural stability of $\text{LiNi}_{0.96}\text{Al}_{0.04}\text{O}_2$ by reducing stress buildup at a high de-lithiation stage and alleviating the crack formation of the secondary particles. Compared to the conventional method used to synthesize $\text{LiNi}_{0.96}\text{Al}_{0.04}\text{O}_2$, the above method provides good rate and cycle performance up to 300 cycles in half-cell with Li metal anode. The method can also successfully employ different oxide seeds (SiO_2) to nickel-rich layered oxides. While this manuscript provides some novel findings and shows solid evidence of the structural mechanism, there are a few suggestions that can be considered to improve the manuscript prior to being accepted for publication in Nature Communications.*

Response: We appreciate the reviewer for the careful review and positive comments on our submitted work. We also thank the reviewer for the thoughtful and constructive suggestions to improve the overall quality of this work, which help us a lot during the revision process of our manuscript. Below, we provide the point-by-point responses to the concerns raised by the reviewer.

1. *It is still unclear how the modified method can give a denser packing density with a smaller particle size. Since the work claims that there is a gradient doping of Al in the secondary particles, why would the dopant affect the particle size to the same degree throughout the particles?*

Response: Thanks for the comments. We agree with the reviewer’s comment that a smaller particle size would typically result in a looser packing density in the accumulation process of the lithium-ion cathode particles. However, the as-obtained nickel-rich layered oxide cathodes (denoted as “hk-LiNi_{0.96}Al_{0.04}O₂”) in our case possess a gradient size distribution of the primary particles because of the gradient doping of Al throughout the secondary particle, with a general trend that the closer to the core, the smaller the primary particles are (**Fig. R1**). We think the reviewer has misunderstood our results, so that we want to emphasize them again. As shown in **Fig. 1i** in the maintext and **Fig. R1** in the response letter, we clearly see that the primary particle size near the core region is much smaller and those near the surface has strikingly larger size with a rod shape. This phenomenon should be attributed to the gradient doping of Al inside of the hk-LiNi_{0.96}Al_{0.04}O₂ secondary particle. Specifically, more Al doping would retard the grain boundary migration of the primary particle, and thus inhibit its growth into larger size.

Fig. R1 Cross-sectional images and the corresponding schematic illustration of the $\text{LiNi}_{0.96}\text{Al}_{0.04}\text{O}_2$ (**a, b**) and $\text{hk-LiNi}_{0.96}\text{Al}_{0.04}\text{O}_2$ (**d, e**). **c, f** Primary particle size distribution of the two samples, as a function of the distance from the center of secondary particle.

As for the concern raised by the reviewer that “It is still unclear how the modified method can give a denser packing density with a smaller particle size”, **we think there are several parameters accounting for this phenomenon.**

The first one is the preferred growth of Ni(OH)₂ precursors on the Al₂O₃ seeds in the heterogeneous nucleation process. When Al₂O₃ oxide seeds are added into the reaction solution, due to the fact that the bonding energy of xNi(OH)₂-(1-x)Al₂O₃ is much stronger than that of Ni(OH)₂-Ni(OH)₂ (**Fig. S2**), the Ni(OH)₂ precursors has a stronger tendency to grow on the alumina rather than forming the individual Ni(OH)₂ particles (**More details can be referred to Research Note II in supporting information**). This nucleation process (**i.e. heterogeneous nucleation process**) is catalyzed by a heterogeneity such as an accommodating substrate surface. While the nucleation to form individual Ni(OH)₂ particles without adding accommodating substrate surface is called **homogeneous nucleation process**. As is discussed in the supporting information, the **energy barriers for heterogeneous nucleation are much lower than those of homogeneous nucleation, resulting in a much faster nucleation rate of heterogeneous process** (*Chem. Mater.* 2009, 21, 1500–1503).

As shown in **Fig. R2a**, the Ni(OH)₂ precursor is tended to form plate-like structure because of the anisotropic growth (*J. Am. Chem. Soc.* **2022**, 144, 20, 8969–8976; *Nat. Commun.* **2020**, 11, 5181; *J. Alloys Compd.* **2015**, 619, 846–853). The plate-like precursors would agglomerate together to grow into a large particle. In a specific co-precipitation time (which includes nucleation time and growth time), since the heterogenous nucleation process has a much faster nucleation rate, it leaves a much

longer growth time than that in the homogenous nucleation process (Fig. R2b). Therefore, the precursors in the heterogeneous nucleation would grow into much thicker and denser plates than those in homogeneous nucleation process. This can be also verified by the SEM images of the precursors in Fig. R2c, 2d and 2e, reflecting that the packing density of the precursors prepared by the reported method is much denser than that of samples prepared by the conventional method (Fig. R2f, 2g and 2h).

Fig. R2 a, Scheme of the anisotropic growth process of the LiNi_{0.96}Al_{0.04}O₂ cathode (*J. Alloys Compd.* **2015**, *619*, 846–853). **b**, Comparison of the nucleation time and growth time of the two different LiNi_{0.96}Al_{0.04}O₂ cathode precursors during the co-precipitation process. **c, d, e**, SEM images of the c-LiNi_{0.96}Al_{0.04}O₂ precursors. **f, g, h**, SEM images of the c-LiNi_{0.96}Al_{0.04}O₂ precursors

The second one is the Kirkendall effect in the calcination process. The hk-precursor shows a distinct core-shell structure with Al₂O₃ seeds located in the core. During the annealing process, the Al atoms will be gradually migrated to the outer surface leaving a gradient doping of Al to the secondary particle with higher concentration of Al at the core region and lower concentration at the outer surface. According to previous literatures, lower Al doping will result in larger primary particle size than those with higher Al doping, as verified by **Fig. R1d, 1e and 1f**. As shown in **Fig. R1e**, the areas with large primary particle are always accompanied with some small primary particle, and this feature also makes the resulting hk-LiNi_{0.96}Al_{0.04}O₂ secondary particle a denser packing density.

To conclude, we think the heterogenous nucleation process and the Kirkendall effect in the annealing process both account for the denser packing density of the hk-LiNi_{0.96}Al_{0.04}O₂ secondary particle. We also sincerely apologize for the confusion to the reviewer that the reviewer thinks “*the dopant affects the particle size to the same degree throughout the particles*”. We have revised the corresponding statements and replaced the corresponding figures in the maintext to avoid further confusions to the readers.

Changes made in the maintext: We have revised **Fig. S7** by adding **Fig. R1** and its corresponding results into a new **Fig. S7**.

2. In general, according to reported research, the smaller grain size of primary particles will increase the active surface area to react or more interfacial reactions with the electrolyte at a high stage of charge. However, this work has a different

discussion that focuses more on the inner bulk of the secondary particles. As the Al dopants from this method locate mostly in the inner bulk, how would this help with the crack formation induced from the surface of the secondary particles? Please explain and include this clarification in the manuscript to avoid any confusion.

Response: Thanks for the comments and suggestions made to our manuscript. We agree with the reviewer's comment that the smaller grain size of primary particles will increase the active surface area to react or more interfacial reactions with the electrolyte at a high stage of charge. According to the reviewer's comment, we have conducted BET measurements on the resulting and control samples in our work. As shown **Fig. R3**, the hk-LiNi_{0.96}Al_{0.04}O₂ secondary particle shows a slightly smaller specific surface area (SSA) (0.19 m²/g) than those of the conventional sample (0.21 m²/g). Considering the secondary particle in both synthetic methods are in spherical shape with similar size, the contribution of the specific surface area should be from the primary particle and interparticle spaces between adjacent primary particles (*Adv. Energy Mater.* **2021**, *11*, 2003400). Thus, the smaller SSA in the hk-LiNi_{0.96}Al_{0.04}O₂ sample reveals that it has less active surface to react with the electrolyte at a high stage of charge.

Fig. R3 Specific surface area of the c-LiNi_{0.96}Al_{0.04}O₂ (a) and hk-LiNi_{0.96}Al_{0.04}O₂ (b) samples.

As for the concern about the crack formation induced from the surface of the secondary particles, we believe that the changes occurred at the surface layers are attributed to the phase transformation from the layered structure to the rock-salt structure, which is a kind of interfacial reaction occurred in the thickness of nanometer scale. This kind of phase transformation is verified to result in negligible crack formation at the surface (*Nano Energy* **2021**, *83*, 105854; *Nat. Commun.* **2020**, *11*, 4433). In real cases, cracks are identified to be more likely generated in the interior region rather than the surface layer. Previous studies by Yuefei Zhang et al. (*ACS Energy Lett.* **2021**, *6*, 1703–1710) and Yang-Kook Sun et al. (*ACS Energy Lett.* **2018**, *3*, 3002–3007) have shown the generation and outward expansion of cracks in the interior region within single secondary particle as the SOC increase. Therefore, many studies have used doping strategy to delay crack propagation (*Chem. Mater.* **2018**, *30*, 1808-1814; *ACS Sustainable Chem. Eng.* **2018**, *6*, 5653-5661), or coating techniques to avoid thorough pulverization of cracked particles (*ACS Appl. Mater. Interfaces* **2018**, *10*, 27821–27830; *ACS Appl. Energy Mater.* **2021**, *4*, 10012-10024; *Adv. Mater.* **2023**, *35*, e2209483).

Fig. R4 a, Scheme of the resulting secondary particle with interior-rich Al dopant and central void structure. **b**, Volume changes as a function of delithiation under different aluminum doping amounts. **c**, Two-phase separation within $hk\text{-LiNi}_{0.96}\text{Al}_{0.04}\text{O}_2$ compared with the control sample during H2-H3 phase transition.

In our case, we developed a unique synthetic strategy to synthesize the secondary particle (nickel content up to 96%) with interior-rich Al dopant and central void structure (**Fig. R4a**) to uniformize the stress distribution. To verify the effect of Al doping to inhibit the structural degradation, we constructed the secondary particles with different Al doping contents, and examined the changes of lattice parameter among these samples. As shown in **Fig. R4b**, the volume changes as a function of delithiation under different aluminum doping amounts (2%, 4% and 6%, respectively) were calculated based on the Bragg equations. In the early stage of the charging process, it can be seen that the three Al-doped samples have a similar evolution trend of volume. Notable difference in the trend appears at the end of charge, where the 6% Al-doped sample exhibit smaller volume changes compared to the low Al-doped samples.

In addition to the volume changes, the H2-H3 two-phase transition behavior also reflects the structural stability of the Nickel-rich layered oxide cathode. As shown in **Fig. 2i** in the maintext, The H2-H3 phase separation occurs at the high charge voltage region, and induces inhomogeneous contraction and structural degradation within particles. In order to highlight the advantages of the proposed method, we show the relative contents of the phase components during the H2-H3 phase separation in the resulting and control $\text{LiNi}_{0.96}\text{Al}_{0.04}\text{O}_2$ samples. As shown in **Fig. R4c**, a stable, continuous evolution of c parameter can be observed in the resulting hk- $\text{LiNi}_{0.96}\text{Al}_{0.04}\text{O}_2$ samples, representing a stable single-phase transition particularly at such high voltage charging process. In sharp contrast, the c- $\text{LiNi}_{0.96}\text{Al}_{0.04}\text{O}_2$ control samples, the c parameter exists an abrupt variation during the SOC ranges from 68% to 80%. This

drastic change in c parameter reflects the structural instability of the control sample, and a plenty of cracks have formed in the secondary particle. The cracks in the secondary particle would induce large loss of conductivity and structural inhomogeneity, which aggravates the stress concentration within secondary particles during H2-H3 phase transition. The as-obtained interior Al-rich structure with enhanced mechanical stability could render better phase-transition homogeneity (**Fig. R4c**) and uniformize the stress distribution in the secondary particle during H2-H3 transition.

In summary, it can be expected that the hk-LiNi_{0.96}Al_{0.04}O₂ sample with a Kirkendall central void structure and a gradient Al doping could effectively alleviate the stress concentration, inhibit the crack formation in the interior structure and maintain the structural integrity of the secondary particles.

3. In Figure 3e, the specific capacity of the c-LiNi_{0.96}Al_{0.04}O₂ sample in full cell suddenly drops at around the 35th cycle. Is the performance repeatable? Usually, it should decrease gradually.

Response: Thanks for bringing up this important point. According to the reviewer's suggestion, we have repeated the full cell experiments, and find that the performance of the c-LiNi_{0.96}Al_{0.04}O₂ is repeatable at some extent. The full cell based on the control c-LiNi_{0.96}Al_{0.04}O₂ sample with the N/P ratio of 1.13 still shows a sudden drop at 30th cycle (red line in **Fig. R5a**). To be noted, the current density we used here is 1C, because we want to accelerate the measurement.

Fig. R5 a, Electrochemical performance of the full cell based on $c\text{-LiNi}_{0.96}\text{Al}_{0.04}\text{O}_2$ samples with various N/P ratio: 1.13, 1.23 and 1.33. Note that: the current density for the test is 1C. **b**, Electrochemical performance of the full cell based on $hk\text{-LiNi}_{0.96}\text{Al}_{0.04}\text{O}_2$ samples with various N/P ratio: 1.05, 1.13 and 1.23.

The capacity fading and retention is closely related to the areal capacity ratio of negative and positive electrodes (N/P ratio). As shown in **Fig. R5**, we have fabricated different full cells based on the resulting $hk\text{-LiNi}_{0.96}\text{Al}_{0.04}\text{O}_2$ and the control $c\text{-LiNi}_{0.96}\text{Al}_{0.04}\text{O}_2$ samples. As for the reason for the sudden drop of the capacity at around 35th cycle, we believe that this observation is generally caused by the inappropriate N/P ratios. When the full cell is assembled with a low N/P ratio, meaning that a low amount of negative electrode, the amount of lithium ion in the system cannot guarantee the normal operation because the formation of CEI and SEI will consume a plenty of electrolyte and lithium ions. Particularly in the $c\text{-LiNi}_{0.96}\text{Al}_{0.04}\text{O}_2$ control samples, the inferior structural stability will cause a lot of Ni species being dissolved in the electrolyte, directly leading to the serious capacity decay (*Adv. Energy Mater.* **2023**, 2302209). The dissolved Ni species in the electrolyte will further react with anode materials (as supported by the TOF-SIMS characterization in **Fig. 4e** and **4f**), causing the instability of the as-formed SEI and further consuming more extra electrolyte. This may be the reason why the capacity suddenly drops in a short-term cycling test. When

the N/P ratio increases to 1.23 and 1.33, the phenomenon of sudden drop in capacity has been improved and the capacity gradually decreases as the cycling number prolongs. Therefore, we replaced the cycling performance of the c-LiNi_{0.96}Al_{0.04}O₂ full cells by using the data based on the new N/P ratio of 1.23, which we think may reflect the more accurate electrochemical performance of the control sample. On the contrary, the resulting hk-LiNi_{0.96}Al_{0.04}O₂ samples, which shows a more stable structure, exhibit negligible Ni dissolution in the electrolyte, consequently delivering a much higher capacity and lower capacity decay rate.

The underlying reason for the effects of N/P ratios on the cycling performance in the full cell is a very important and complicated topic. However, the underlying reason is not directly related to the findings we report in this work. In order to not dilute the focus and highlight of our work, we think it can be a systematic and independent study which needs substantial work to reveal. We hope the reviewer will follow our next work on this topic.

Changes made in the maintext: We have revised **Fig. 3e** by replacing cycling performance of the full cell based on the c-LiNi_{0.96}Al_{0.04}O₂ cathode and graphite anode with a N/P ratio of 1.23.

4. There are several grammatical errors and typos, please revise the writing more thoroughly. For instance, page 11 has repetitive sentences that can be put in the previous paragraph (Simulation of the stress in secondary particles part). On page 17, Figure S12 needs to be changed to Figure S11 (Improved electrochemical performance by the uniform stress distribution design part). On page 22, it should be secondary

particles instead of second particles (conclusion part).

Response: Thanks for noting this and we have corrected relevant errors.

Changes made in the maintext: We have revised the simulation details of the stress in the secondary particles by using new data. We have thoroughly double checked this part, and we believe there are no grammatical errors and typos in the revised manuscript. We have also revised the typos accordingly.

5. One of the main issues for nickel-rich layered oxides is related to thermostability issues at a high stage of charge. It would be good to have a gas release test or DSC data of the samples prepared by the modified method compared to the conventional method.

Response: Thanks for this important suggestion. Thermostability of the resulting cathode material at a high stage of charge is a very critical parameter when considering its practical applications. According to the reviewer's suggestion, we have conducted the differential scanning calorimetry (DSC) test and operando time-resolved (TR)-XRD experiments (150–400 °C) using delithiated cathodes (cut-off at 4.3 V vs Li/Li⁺).

DSC profiles of the two samples (**Fig. R6a**) show the higher thermal stability of *hk*-LiNi_{0.96}Al_{0.04}O₂. The *c*-LiNi_{0.96}Al_{0.04}O₂, which has the same Ni content (96%) as that of *hk*-LiNi_{0.96}Al_{0.04}O₂, delivers a notable peak temperature at 215 °C. For *hk*-LiNi_{0.96}Al_{0.04}O₂, the peak appears at 224 °C, and the enthalpy quantification of the exothermic peak is 1743.37J g⁻¹, which is much lower than that of *c*-LiNi_{0.96}Al_{0.04}O₂ (2412.85 J g⁻¹). This implies less oxygen release during the structure collapse near the

material surface. During the heating, the layer-structured hk -LiNi_{0.96}Al_{0.04}O₂ completely transform into a spinel structure at 250 °C (**Fig. R6b and 6c**), higher than c -LiNi_{0.96}Al_{0.04}O₂ (220°C), as evidenced by merging of the (108)R and (110)R peaks into a single peak attributed to spinel (440)S. Then both the two disordered spinel phase samples start to transform into rock salt structure (Fm $\bar{3}$ m) at 320 °C. Less interfacial exposure in the charging state could explain these observations in the thermal stability test. As a result, the hk -LiNi_{0.96}Al_{0.04}O₂ samples exhibit higher thermal stability than c -LiNi_{0.96}Al_{0.04}O₂.

Fig. R6 Thermal stability. **a** DSC profile of c -LiNi_{0.96}Al_{0.04}O₂ and hk -LiNi_{0.96}Al_{0.04}O₂ (cut-off at 4.3 V vs Li/Li⁺) **b, c** XRD of the delithiated (cut-off at 4.3 V vs Li/Li⁺) c -LiNi_{0.96}Al_{0.04}O₂ and hk -LiNi_{0.96}Al_{0.04}O₂ during in-situ heating.

Changes made in the maintext: According to the reviewer's suggestion, we have added **Fig. R6** into supporting information as a new **Fig. S18**.

Point-to-point response to Reviewer #2:

General comments: The manuscript titled “Kirkendall effect induced uniform stress distribution stabilizes nickel-rich layered oxide cathodes” presents a strategy to uniformize the stress distribution in secondary particle facilitated by Kirkendall effect during the fabrication process to stabilize the high-Ni cathode during electrochemical cycling. As proof of the proposed concept, the authors employed Al₂O₃ as the "seed" to fabricate the yolk-like structured cathode precursor. Subsequently, the calcination process generated a dopant-rich interior structure with a central Kirkendall void. The authors also conducted DFT studies to support this proposition. However, this study lacks the urgency and novelty required for publication in the top journal as Nature Communications. Following some revisions suggested below, it would be more suitable to submit this work to Communications Chemistry for potential publication.

Response: We appreciate the reviewer for the careful review on our submitted work. The reviewer concerns about this study lacks the urgency and novelty required for publication in the top journal as *Nature Communications*. The reviewer thinks that the Kirkendall effect in our work is not novel because this method has been used to synthesize the hollow structure as lithium ion battery electrodes. However, we debate our method is not same with the work listed by the reviewer, and there are many highlights in our work, for instance, the new composition of the Nickel-rich cathode, the novel features brought by the Kirkendall effect and the impressive performance of the resulting samples, all of which are not accomplished by the work listed by the reviewer. Another concern about this work which the reviewer thinks the work lack

urgency is the reported performance. To eliminate the reviewer's concern, we have re-examined the electrochemical performance of the resulting and control samples and updated the data in the maintext and supporting information. We believe our updated results have gained much improvement and satisfy the urgency requirement for the prestigious journal like *Nature Communications*. Below, we provide the point-by-point responses to the concerns raised by the reviewer.

1. *The Kirkendall effect has been utilized during the cathode fabrication process to facilitate the generation of primary particles with hollow cores since 2012 (see Angew. Chem. Int. Ed. 2012, 51, 239–241.). The resulting void structures did enhance the electrochemical stability of the cathode to a certain extent (see Metal segregation in hierarchically structured cathode materials for high-energy lithium batteries Nat. Energy 1, 15004 (2016)). The difference of this study: employing a modified protocol, using Al₂O₃ and silica to form a yolk-like structured cathode precursor to enable the dopant-rich interior structure with a central Kirkendall void. The resulting cathode h_k-LiNi_{0.96}Al_{0.04}O₂ delivers an improved cycling stability but not impressive (especially for the rate performance in Fig. 3a and the cycling performance of Li-ion cells in Fig. 3e) to ensure its publication in top journal as Nature Communications. Unprecedented cycling performance is essential for validating a modified or new strategy/concept.*

Response: Thanks for the comments and noting these valuable references. We have carefully read these references mentioned by the reviewer, and we think the main findings of these papers are totally different from ours, although the *Angew* paper reported the usage of Kirkendall effect to synthesize the hollow electrode materials and

the *Nature Materials* paper elucidates the metal segregation (particular Ni metal) on the NCM cathode. More specifically, the *Angew* paper reported by Lou et al. synthesized the hollow $\text{LiNi}_{0.5}\text{Mn}_{1.5}\text{O}_4$ cathode with excellent rate capability and cycling stability. The authors emphasized the enhanced ion transport kinetics brought by the hollow structure prepared by the Kirkendall Effect. The *Nature Materials* paper by Doeff and Xin et al., reported the usage of advanced characterization techniques to reveal the elemental segregation in $\text{LiNi}_{0.4}\text{Mn}_{0.4}\text{Co}_{0.2}\text{O}_2$ spherical particles, which could have positive effect on inhibiting the surface reconstruction on the NMC cathode. The resulting electrode displays improved high-voltage cycling performance majorly owing to the surface chemistry control via the metal segregation phenomenon. The highlights of these two papers are different from the findings in our work. To the best of our knowledge there are currently no study employing this classical concept-Kirkendall effect-to address the **severe structural degradation issues of ultra-high nickel cathode materials**. Based on the evaluation of the material property and impressive performance of the resulting cathodes, we believe that our work could reach the high novelty requirements of the prestigious *Nature Communications* journal. Herein, we want to emphasize the novelty and highlights our work again as listed below. **(1) new compositional and structural design strategy:** we apply different oxide doping seeds in the synthetic procedures to fabricate different compositional ultra-high nickel layered oxide cathodes ($\text{LiNi}_{0.96}\text{Al}_{0.04}\text{O}_2$, $\text{LiNi}_{0.8}\text{Co}_{0.15}\text{Al}_{0.05}\text{O}_2$ and $\text{LiNi}_{0.98}\text{Si}_{0.02}\text{O}_2$) with central Kirkendall void structure and gradient metal doping. Nickel-rich cathode materials with such compositions and structure have not been reported in the literatures. **(2) New**

stabilization strategy focusing on the core region within the secondary particle:

induced by the novel Kirkendall void structure and the interior Al-rich doping, the stress distribution of the secondary particle can be uniformized, thus resulting in a stabilized crystal structure and maintaining the structural integrity during the electrochemical cycling processes. **(3) Impressive performance enhancement brought by the structural design:** the above compositional and structural features endow superior electrochemical properties and much enhanced rate and cycling performance, which outperform the state-of-the-art ultrahigh nickel layered oxide cathodes.

Fig. R7 Electrochemical performance re-test of $c\text{-LiNi}_{0.96}\text{Al}_{0.04}\text{O}_2$ and $hk\text{-LiNi}_{0.96}\text{Al}_{0.04}\text{O}_2$. **a**

Cycling stability of half-cells at high cut-off voltages and more practical areal loading (about

10 mg/cm²). **b** Rate performance of half-cells under more practical areal loading (about 10

mg/cm²) at 0.1C, 0.2C, 0.5C, 1C, 2C and 5C respectively. **c** Cycle performance of coin-type full-cells with graphite anodes between 2.7 and 4.6V at 0.3 C.

As for the other concern about the electrochemical performance raised by the reviewer, we have made attempts to modify the synthesis parameters to further improve the electrochemical performance. We find that the sulfate, involved in the coprecipitation, remained in the precursor even under cold water washing. The general calcination temperature (below 800 °C) makes it difficult for SO₄²⁻ to decompose completely, and thus leads to the deterioration of electrochemical performance. We have re-washed the precursor with hot sodium carbonate solution and deionized water to solve this issue.

We re-evaluate the cycling performance and rate capability of the two samples under high cut-off voltage (4.6 V) and practical load (> 10 mg/cm²) conditions (**Fig. R7a**). At high cut-off voltages, the deep extraction of lithium ions in the lattice would induce more drastic lattice contraction, thereby severe volume change, which is a challenge to the mechanical stability of the material. When charged to 4.6 V, the c-LiNi_{0.96}Al_{0.04}O₂ shows a specific capacity of 226 mAh/g at a current density of 0.3C. Because of the large volume change at such a high cut-off voltage, the control cathode displays a serious capacity decay with a specific capacity of 163 mAh/g after 120 cycles and a large capacity decay rate of 0.2%/cycle. The inferior structural property of the c-LiNi_{0.96}Al_{0.04}O₂ cathode also results in a bad rate capability (**Fig. R7b**). Specifically, the c-LiNi_{0.96}Al_{0.04}O₂ cathode delivers 235 mAh/g, 216 mAh/g, 188 mAh/g, 170 mAh/g and 147 mAh/g at 0.1C, 0.2C, 0.5C, 1C and 2C, respectively. When the current density

increasing to 5C, the capacity quickly drops to 63 mAh/g, which should be attributed to the crack formation and serious polarization in the c-LiNi_{0.96}Al_{0.04}O₂ sample.

On the contrary, the unique structure design in resulting hk-LiNi_{0.96}Al_{0.04}O₂ sample could effectively alleviate the crack formation and material polarization at deep charging condition. As a result, the resulting cathode shows a high specific capacity of 225 mAh/g at 0.3C and good cycling stability, still delivering a high specific capacity of 200 mAh/g after 150 cycles with a low decay rate of 0.07%/cycle. In addition, the unique structural property of the resulting hk-LiNi_{0.96}Al_{0.04}O₂ sample also possesses good rate capability, exhibiting 234 mAh/g, 230 mAh/g, 225 mAh/g, 210 mAh/g and 187 mAh/g at 0.1C, 0.2C, 0.5C, 1C and 2C, respectively. The resulting cathode can sustain the structural stability when the current density reaches 5C, with a high specific capacity of 150 mAh/g.

We also supplement the cycle performance of the full-cells, of which the N/P ratio was carefully controlled (**Fig. R7c**). In the case of c-LiNi_{0.96}Al_{0.04}O₂, a severe capacity decay can be observed upon the early cycle, where the capacity retention over 100 cycles was recorded as 69%. In the full-cell system with limited lithium source, the interfacial reaction caused by cracks mostly contribute to the consumption of available lithium, leading to rapid capacity decay during cycle. In contrast, hk-LiNi_{0.96}Al_{0.04}O₂ exhibits a high initial capacity (208 mAh·g⁻¹) at 1C (180 mAh·g⁻¹), retaining a retention of 86% over 500 cycles with an average capacity loss of 0.028%/cycle. The depression of cracks formation reduces the exposed fresh surface area during repetitious volume change of cathode material upon cycle. The enhanced mechanical stability lead to less

irreversible consumption of Li^+ and transition metal dissolution (Ni). As a result, the $\text{hk-LiNi}_{0.96}\text{Al}_{0.04}\text{O}_2$ exhibits impressive electrochemical stability during long-term cycling.

We believe our updated results have gained much improvement, and these results suggest uniformizing the stress distribution in the secondary particle represent a novel strategy to enable high energy density and long cycle life in nickel-rich layered oxide cathodes. To conclude, we believe that our paper and results possess a high level of novelty and satisfy the urgency requirement for the prestigious journal like *Nature Communications*.

2. The electrochemical testing and characterization of the resulting cathode $\text{hk-LiNi}_{0.98}\text{Si}_{0.02}\text{O}_2$ are missing, the authors only present the core structure of the precursor rather than present the $\text{hk-LiNi}_{0.98}\text{Si}_{0.02}\text{O}_2$. This compromises the integrity of the work and the range of the developed method to some extent. It is highly recommended to focus on presenting the experimental data rather than speculating on the potential of the current approach.

Response: Thanks for the valuable suggestions. We agree with the reviewer that the supplement of the electrochemical testing and characterization of the resulting $\text{hk-LiNi}_{0.98}\text{Si}_{0.02}\text{O}_2$ and $\text{hk-LiNi}_{0.80}\text{Co}_{0.15}\text{Al}_{0.05}\text{O}_2$ cathode will guarantee the integrity of the work and the range of the developed method to some extent. We explored the internal morphology (**Fig. R8a**) and element distribution of Si within $\text{LiNi}_{0.98}\text{Si}_{0.02}\text{O}_2$ particles, where the point scan analysis results show the much higher signal of Si near the Kirkendall void (spot #1) formed by high-temperature diffusion in the EDS test (**Fig.**

R8b). According to the reviewer's suggestion, we have measured the electrochemical performance and conducted the performance comparison between the resulting and control $\text{LiNi}_{0.98}\text{Si}_{0.02}\text{O}_2$ as well as $\text{LiNi}_{0.80}\text{Co}_{0.15}\text{Al}_{0.05}\text{O}_2$ samples. As shown in **Fig. R9**, the hk- $\text{LiNi}_{0.98}\text{Si}_{0.02}\text{O}_2$ delivers much better capacity at various C rates, indicating its less material polarization and superior rate capability. When coupled with the conventional graphite anode with the N/P ratio of 1.13, the full cell based on the hk- $\text{LiNi}_{0.80}\text{Co}_{0.15}\text{Al}_{0.05}\text{O}_2$ also exhibits much better cycling stability compared to that based on the control samples. These results can verify the advantages brought by the reported structural design strategy, and its excellent diversity to obtain other nickel-rich layered oxide cathode materials.

Fig. R8 a The cross-sectional SEM image of $\text{hk-LiNi}_{0.98}\text{Si}_{0.02}\text{O}_2$ secondary particle. **b** EDS point-scan spectra of $\text{hk-LiNi}_{0.98}\text{Si}_{0.02}\text{O}_2$ at three different positions.

Fig. R9 Electrochemical performance of the as-obtained $hk\text{-LiNi}_{0.98}\text{Si}_{0.02}\text{O}_2$ and $hk\text{-LiNi}_{0.85}\text{Co}_{0.15}\text{Al}_{0.05}\text{O}_2$ cathodes. **a** Cycling stability of half-cells at high cut-off voltages and more practical areal loading (about 10 mg/cm^2) of $c\text{-LiNi}_{0.98}\text{Si}_{0.02}\text{O}_2$ and $hk\text{-LiNi}_{0.98}\text{Si}_{0.02}\text{O}_2$ at 0.3C. **b** Charge/discharge curves of the $hk\text{-LiNi}_{0.98}\text{Si}_{0.02}\text{O}_2$ cathode at 0.3C for 100 cycles. **c** The charge and discharge mid-voltage of half-cells for $c\text{-LiNi}_{0.98}\text{Si}_{0.02}\text{O}_2$ and $hk\text{-LiNi}_{0.98}\text{Si}_{0.02}\text{O}_2$ during cycle. **d** Rate performance of half-cells under more practical areal loading (about 10 mg/cm^2) of $c\text{-LiNi}_{0.98}\text{Si}_{0.02}\text{O}_2$ and $hk\text{-LiNi}_{0.98}\text{Si}_{0.02}\text{O}_2$ at 0.1C, 0.2C, 0.5C, 1C, 2C, 5C and 7C respectively. **e** Cycling stability of full-cells of $c\text{-LiNi}_{0.85}\text{Co}_{0.15}\text{Al}_{0.05}\text{O}_2$ and $hk\text{-LiNi}_{0.85}\text{Co}_{0.15}\text{Al}_{0.05}\text{O}_2$ at 0.5C. **f** Rate performance at 0.1C, 0.2C, 0.5C, 1C, 2C and 5C respectively.

Changes made in the maintext: We have revised **Fig. 3** (the electrochemical performance results) by using the new electrochemical data of the resulting and control cathode samples measured during the revision process. We have also re-written the corresponding analysis for these as-obtained results.

3. There is an issue with the description of "In the specific synthetic procedure, EM oxide microparticles such as alumina (Al_2O_3) and silica (SiO_2) were introduced to the co-precipitation system and acted as the seed for the heterogeneous nucleation of nickel hydroxide precursors." It is evident that silica is not an exotic metal (EM) oxide. The authors need to correct this and ensure accurate information is conveyed.

Response: We really appreciate the reviewer for providing this valuable suggestion to avoid potential confusion to the readers. We admit that the usage of exotic metal oxide to refer Al_2O_3 and SiO_2 is inappropriate because SiO_2 technically is not a metal oxide. After thorough considerations, we decide to revise the "exotic metal oxide" into "exotic metal/metalloid oxide". Thus, we think our revised description will cover both Al_2O_3 and SiO_2 .

Changes made in the maintext: We have revised the word of "exotic metal oxide" into "exotic metal/metalloid oxide" throughout the paper.

4. The caption of Figure 21 is confusing and should be corrected as "Figure S21 a Cross-sectional SEM image and b corresponding element mapping of the Al_2O_3 nucleus introduced into $Ni_{0.8}Co_{0.15}(OH)_2$ to form the $LiNi_{0.8}Co_{0.15}Al_{0.05}O_2$ precursor. c, b SEM images and cross-sectional morphology of the precursors for NCA cathode materials obtained by routine co-precipitation method under the same synthesis conditions."

Response: Thanks for the correction. The caption has been corrected and highlighted in the revised supporting information.

Changes made in the maintext: We appreciate the reviewer's suggestion, have revised the caption of **Fig. S21**.

Point-to-point response to Reviewer #3:

Remarks to the authors:

This work by Li et al. talks about using Kirkendall effect as a tool to optimize the stress distribution in the $\text{LiNi}_{0.96}\text{Al}_{0.04}\text{O}_2$ therefore improve the cycling stability of cathodes.

This paper is potentially useful but filled with problems that need to be addressed before being accepted in Nature Communications.

Response: We appreciate the reviewer for the careful review and positive comments on our submitted work. We also thank the reviewer for the thoughtful and constructive suggestions to improve the overall quality of the work, which help us a lot during the revision process of our manuscript. Below, we provide the point-by-point responses to the concerns raised by the reviewer.

Here are my specific complaints.

1. *As being said by the authors, the introduction of Al was reported to decrease the primary particle size, why does the hk- $\text{LiNi}_{0.96}\text{Al}_{0.04}\text{O}_2$ has smaller particle size on the surface than c- $\text{LiNi}_{0.96}\text{Al}_{0.04}\text{O}_2$, where c- $\text{LiNi}_{0.96}\text{Al}_{0.04}\text{O}_2$ is expected to have higher Al concentrations on the surface?*

Response: Thank you for your valuable comments. The statement of “The introduction of excess Al was reported to limit the primary particle size” specifically refers to that the size of the primary particle at the hk- $\text{LiNi}_{0.96}\text{Al}_{0.04}\text{O}_2$ core region is relatively smaller than that in the c- $\text{LiNi}_{0.96}\text{Al}_{0.04}\text{O}_2$ control sample. This can be clearly identified by the cross-section SEM and its schematic illustration (**Fig. R1a and 1b**). The feature of smaller size will benefit the intimate contact between primary particles, facilitating the

charge transport on the interface, and thus reducing the mechanical pulverization and homogenizing the charge distribution within secondary particle.

Fig. R1 Cross-sectional images and the corresponding schematic illustration of the *c*-LiNi_{0.96}Al_{0.04}O₂ (**a, b**) and *hk*-LiNi_{0.96}Al_{0.04}O₂ (**d, e**). **c, f** Primary particle size distribution of the two samples, as a function of the distance from the center of secondary particle.

As for the reviewer's concern that "why does the *hk*-LiNi_{0.96}Al_{0.04}O₂ has smaller particle size on the surface than *c*-LiNi_{0.96}Al_{0.04}O₂, where *c*-LiNi_{0.96}Al_{0.04}O₂ is expected to have higher Al concentrations on the surface?", we have carried out SEM characterizations on the cross section of the resulting *hk*-LiNi_{0.96}Al_{0.04}O₂ and the *c*-LiNi_{0.96}Al_{0.04}O₂ control samples. We can clearly see that the primary particle size near the core region is much smaller and those near the surface has strikingly larger size with a rod shape (**Fig. R1a and 1b**). We also calculate the areas of the primary particles and plot the profile of the area distribution as a function of the distance from the secondary particle center (**Fig. R1c**). It is obvious that the primary particles near the outer surface have a larger average size than those near the core region. In comparison, the cross-section SEM image of the *c*-LiNi_{0.96}Al_{0.04}O₂ control sample (**Fig. R1d**) and its schematic illustration (**Fig. R1e**) which indicate the distribution of primary particle

indicate that there is no specific trend for the size distribution either near the core or near the outer surface. We can also see the primary particles in c-LiNi_{0.96}Al_{0.04}O₂ control sample possess a relatively uniform size distribution in **Fig. R1f**. These results are clearly consistent with the statement we made in the maintext and the general conclusion which has been made from other literatures.

However, we provide the zoom-in SEM images of the resulting hk-LiNi_{0.96}Al_{0.04}O₂ and the c-LiNi_{0.96}Al_{0.04}O₂ control samples in **Fig. S7**, which showing a contradictory result with previous discussion. **We think this may be the key point that the reviewer is concerned about.** We want to make some clarifications here that the size distribution in **Fig. S7c** and **S7d** refers to **the thickness of the LiNi_{0.96}Al_{0.04}O₂**. Since the LiNiO₂ owns a layered structure with the space group of R-3m, it typically forms a plate-like structure after high-temperature calcinating the hydroxide precursors. We are sorry for this confusion, and therefore we have made it clearer in the caption of **Fig. S7** in the supporting information.

Fig. R10 Scheme of the precursors obtained in the reported synthetic method (a) and the conventional co-precipitation method (b).

As for the reasons for this abnormal phenomenon in **Fig. S7**, we believe it is related to the coprecipitation reaction condition. In our reported synthetic method, we adopt Al₂O₃ as the heterogeneous growth seed, and consequently the resulting precursor

grows into a core-shell structure with all of the Al element located in the core region (**Fig. R10a**). While in the coprecipitation process of the control precursor, a proper amount of aluminum isopropoxide in isopropanol is used as the Al source and added into the basic reaction solution. This condition results in the Ni(OH)₂ precursors with uniformly mixed with the Al-dopants (**Fig. R10b**). These two precursors with different microstructures would have different growth behavior during the high-temperature calcination, thus leading to the different morphology of the primary particles in the different LiNi_{0.96}Al_{0.04}O₂ samples, particular for those near the outer surface (c-LiNi_{0.96}Al_{0.04}O₂: random shape; hk-LiNi_{0.96}Al_{0.04}O₂: long rod-like shape). But if we use the volume (estimated by area · thickness) to describe the size of the primary particle, we can still get the consistent results. For example, the average volume of the primary particle near outer surface in c-LiNi_{0.96}Al_{0.04}O₂ is estimated to be 0.0054 μm³ (0.03 μm² · 0.18 μm), while the volume of the corresponding particles in hk-LiNi_{0.96}Al_{0.04}O₂ is estimated to be 0.0072 μm³ (0.06 μm² · 0.12 μm). This estimation also verifies the statement in the maintext and the general conclusion published elsewhere.

Changes made in the maintext: We have revised **Fig. S7** by adding **Fig. R1** and its corresponding results into a new **Fig. S7**. We have also re-written the caption of the new **Fig. S7**.

2. From the strain-stress modeling in Figure 2a, I do not see more strains in the center of the particle, this contradicts the explanation in the paper, and in this case, how does model ii and model iii actually help with releasing the strain? Another concern is how primary particle size would have impact on cracking and surface parasitic reactions,

also how the strain will actually look like when particle size is decreased, and more grain boundary presents?

Response: Thanks for your valuable comments and professional questions. After revisiting the simulation model and results, we find a flaw in our previous model, which is that the simulation models are not constructed based on resulting and control cathode secondary particles. In our revised manuscript, we constructed the simulation model which is closer to the realistic case, by using the CAD to depict the primary particles in the secondary particles according to the corresponding cross-section SEM images.

Fig. R11 Calculated stress distribution as a function of state of charge inside (a) c-LiNi_{0.96}Al_{0.04}O₂ and (b) hk-LiNi_{0.96}Al_{0.04}O₂.

Based on the theory of diffusion induced stress, we have carried out the stress simulation of the two models in COMSOL Multiphysics®. The Young 's modulus,

Poisson's ratio, stress tensor and Kronecker symbol of active particles were referenced from previous study (*Mater. Today* **2020**, *36*, 73-82). The calculated stress distribution as a function of state of charge in the c-LiNi_{0.96}Al_{0.04}O₂ and hk-LiNi_{0.96}Al_{0.04}O₂ is plotted and presented in **Fig. R11**. At a median charge depth (0.5), we can see the primary particles in the c-LiNi_{0.96}Al_{0.04}O₂ control sample already undergo different degrees of stressing and contraction. The stress is mainly distributed between the grains. These red dots and lines (marked by black arrow) in **Fig. R11a** ($x = 0.5$) indicate the areas with very large tensile strength, one can easily find that those high stressed areas are located majorly in the core region of the secondary particle. Due to the anisotropic shrinkage of the primary particles, the large tensile strength near the grains will intensify the mutual extrusion of the adjacent primary particles, leading to the fast growth and formation of cracks in the center of the secondary particle. This result is consistent with the conclusion made in the manuscript and literatures published elsewhere (*ACS Energy Lett.* **2021**, *6*, 1703–1710; *ACS Energy Lett.* **2020**, *6*, 216-223). With the increase of the charge depth, the tensile strength inside the secondary particle keeps increasing and those high stressed areas have transferred to the primary particles at the core region.

In sharp contrast, the tensile and compressive strength distribution of primary particles in the resulting hk-LiNi_{0.96}Al_{0.04}O₂ sample is much more uniform, either at a relatively low charge depth (0.5) or even at a very deep charge depth (0.8). The smaller grain near the Kirkendall void structure in hk-LiNi_{0.96}Al_{0.04}O₂ sample could weaken the directionality of the shrinkage caused by lithium extraction, and further uniformize the

stress distribution inside of the secondary particle. It is worth noting that the average stress in the hk-LiNi_{0.96}Al_{0.04}O₂ sample model with a high charge depth of 0.8 is even much smaller than that in the control sample with a low charge depth of 0.5. All of these simulation results verify the structural features brought by the unique Kirkendall void design and gradient Al doping strategy.

In addition, the reviewer also concerns about the effect of grain size on the crack formation. As we can see from the simulation models for the a-LiNi_{0.96}Al_{0.04}O₂ and hk-LiNi_{0.96}Al_{0.04}O₂ sample, the primary particle in the resulting hk-LiNi_{0.96}Al_{0.04}O₂ sample has smaller size particularly near the core region. The simulation results suggest that the hk-LiNi_{0.96}Al_{0.04}O₂ secondary particle with smaller grain size would suffer from much lower tensile and compressive strength. This was rationalized by the smaller size of the primary grains that enhanced the isotropy of the stress in the polycrystalline secondary particles, leading to better transmission and uniformity of stress. Therefore, the much uniform stress distribution in the resulting cathode could retard the crack formation and inhibit the surface parasitic reactions.

Fig. R12 a The nano-indentation experiment with nano-indentation tester (MDTC-EQ-M55-01) **b, c** An image captured the nanoindentation approach before and after mechanical failure of the representative $c\text{-LiNi}_{0.96}\text{Al}_{0.04}\text{O}_2$ and $hk\text{-LiNi}_{0.96}\text{Al}_{0.04}\text{O}_2$ particle. **d** The corresponding force-displacement curves for indentation phases of the two samples.

We attempt to experimentally evaluate the mechanical strength of the single secondary particles of the resulting and control cathode materials in the nano-indentation test (**Fig. R12**). We believe the results can be used as an indirect proof for the statement that the resulting $hk\text{-LiNi}_{0.96}\text{Al}_{0.04}\text{O}_2$ samples possess better mechanical strength. The force-displacement curves are plotted in **Fig. R12d** to compare the robustness of the secondary particle for $c\text{-LiNi}_{0.96}\text{Al}_{0.04}\text{O}_2$ and $hk\text{-LiNi}_{0.96}\text{Al}_{0.04}\text{O}_2$. With the increase of displacement, the load on the particles would gradually increase until the particle is crushed. It can be seen that the $hk\text{-LiNi}_{0.96}\text{Al}_{0.04}\text{O}_2$ sample is able to bear greater load and deformation before crushing, indicating that the resulting cathode with decreased grain size and interior Al-rich doping owns better mechanical stability than the control sample.

Changes made in the maintext: We have revised the simulation details of the stress in the secondary particles by using new data. In order to keep the fluent logic of the manuscript, we decide to split the simulation results in **Fig. 2** into two figures. The **Fig. R11** in the response letter has been added into the maintext as the new **Fig. 2**. The previous version of **Fig. 2** will be the new **Fig. 3**.

3. What is the C-rate being used for the in-situ XRD experiment, is the c-rate slow enough to resolve the peak splitting due to kinetic limitation of phase transition? The

OCV for c-LiNi_{0.96}Al_{0.04}O₂ is slightly higher than hk-LiNi_{0.96}Al_{0.04}O₂, any explanation for the reason?

Response: Thank you for your valuable comments. It is indeed critical to employ a low enough current density to resolve the peak splitting due to the kinetics limitation of phase transition in the LiNi_{0.96}Al_{0.04}O₂ sample.

Generally, the (001) peaks splitting representing two different sets of lattice parameters are non-overlapping for discontinuous two-phase transitions. The signal intensity of (001) peak is adequate to facilitate us resolve the splitting of the peaks. From the practical experience under the previous experimental conditions, we found that a relatively complete phase separation process can be observed when controlling the current density below 0.2C and the X-ray scan rate below 3°/min. To show more complete peak shift during phase transition process, we have employed a smaller current density (0.08C) to repeat the experiments on the new equipment with stronger signal detection sensitivity in the revision process of the manuscript. As shown in **Fig. R13**, the obtained data plotted in the planar image demonstrates the (dis)continuous shift of peaks position of the two sample more clearly. And the obtained results also have good reproducibility with that in the first draft.

Fig. R13 a Voltage curves of $c\text{-LiNi}_{0.96}\text{Al}_{0.04}\text{O}_2$ (black) and $hk\text{-LiNi}_{0.96}\text{Al}_{0.04}\text{O}_2$ (red) during the operando XRD at the current density of 0.08C. Contour plot of the diffraction patterns showing the (003), (104) and (108)/(110) reflections during charge in the $c\text{-LiNi}_{0.96}\text{Al}_{0.04}\text{O}_2$ (b) and $hk\text{-LiNi}_{0.96}\text{Al}_{0.04}\text{O}_2$ (c).

As for the concern about “*The OCV for $c\text{-LiNi}_{0.96}\text{Al}_{0.04}\text{O}_2$ is slightly higher than $hk\text{-LiNi}_{0.96}\text{Al}_{0.04}\text{O}_2$* ”, we believe the value of OCV may fluctuate within a certain range among these two samples. This is a normal phenomenon and many systematic factors, such as assembly process (particularly pressure), slurry preparation and atmosphere factors, will affect the OCV value. If the reviewer refers to the charging voltage plateau of the $c\text{-LiNi}_{0.96}\text{Al}_{0.04}\text{O}_2$ (black line in **Fig. R13a**) is slightly higher than that of $hk\text{-LiNi}_{0.96}\text{Al}_{0.04}\text{O}_2$ (red line in **Fig. R13a**), the reason for this phenomenon is because the polarization behavior induced by the different structural properties of these two cathode samples.

4. Middle panel of in-situ XRD in Figure 2 should be 101 peak, while the authors say 104, the author should show the change of 104 as well since it should also split at high SOC. There are many misleading labels of Figures in this manuscript especially in the

supporting information, for example, Fig S24, which doesn't exist, Table S2, Fig S12 etc. Many labels should be fixed.

Response: We apologize for the grammatical errors in the first draft, and we have done a thorough check in this revised draft to avoid misleading the readers. Thanks for your careful checks.

Changes made in the maintext: We have carefully double checked the maintext and supporting information, and have revised the paper accordingly.

5. Why the failures for half cells and full cells are different, in detail, half cells for c-LiNi_{0.96}Al_{0.04}O₂ show gradual capacity loss while full cells show a sudden capacity drop after 40 cycles.

Response: Thanks for this important comment. The capacity fading and retention either in the half cell and full cell is closely related to the amount of lithium source, or the areal capacity ratio of negative and positive electrodes (N/P ratio), because the formation of CEI and SEI, as well as the side reaction induced from the metal ion dissolution from the cathode will continuously consume a plenty of electrolyte. As in the half cells for c-LiNi_{0.96}Al_{0.04}O₂ which use lithium metal as the anode, the lithium source is far enough for such continuous consumption. So, the gradual capacity loss in the half cells should be attributed to the material properties (**like structural instability, large amount of Ni ions dissolution in the electrolyte, instability of the SEI...etc. as discussed in the maintext**) of the c-LiNi_{0.96}Al_{0.04}O₂ cathode itself.

However, the full cell based on the c-LiNi_{0.96}Al_{0.04}O₂ cathode is assembled with a specific N/P ratio (1.13), and shows a sudden capacity drop after 40 cycles. We

speculate that the amount of lithium ion in the full cell system cannot guarantee the normal operation because the formation of CEI and SEI will consume a plenty of electrolyte and lithium ions. Particularly in the c-LiNi_{0.96}Al_{0.04}O₂ control samples, the inferior structural stability will cause a lot of Ni species being dissolved in the electrolyte, directly leading to the serious capacity decay. The dissolved Ni species in the electrolyte will further react with anode materials (as supported by the TOF-SIMS characterization in **Fig. 4e** and **4f**), causing the instability of the as-formed SEI and further consuming more extra electrolyte. This may be the reason why the capacity suddenly drops in a short-term cycling test. During our revision process of the manuscript, we increase the N/P ratio to 1.23 and 1.33 to re-assemble the full cells and re-test the electrochemical performance, and the phenomenon of sudden capacity drop in the new full cell devices has been improved and the capacity gradually decreases as the cycling number prolongs. Therefore, we replaced the cycling performance of the c-LiNi_{0.96}Al_{0.04}O₂ full cells by using the data based on the new N/P ratio of 1.23, which we think may reflect the more accurate electrochemical performance of the control sample.

Changes made in the maintext: We have revised **Fig. 3e** by replacing cycling performance of the full cell based on the c-LiNi_{0.96}Al_{0.04}O₂ cathode and graphite anode with a N/P ratio of 1.23.

6. I believe in industry, people can make good NCA by making NC first and add Al in a solid-state synthesis, can you comment on why Al will form a gradient in your case but not for NCA?

Response: We gratefully appreciate for your valuable suggestion. We totally agree with the reviewer's opinion that in industry, people can make good NCA by making NC precursors first and add Al source (typically $\text{Al}(\text{OH})_3/\text{Al}_2\text{O}_3$) in a solid-state synthesis. It seems that the synthetic procedure is very feasible and it is very easy to obtain the NCA samples. However, there is a practical challenge for this method, in which an inert layer of LiAlO_2 or Li_5AlO_4 will be easily formed on the surface of the precursors at the beginning of the calcination process (*J. Mater. Chem. A.* **2015**, 3, 894–904) (*Electrochem. Acta.* **2018**, 291, 84–94). Therefore, higher calcination temperature and longer calcination time are required to fully diffusion Al to achieve the homogeneous phase of NCA.

In our proposed method, we adopt the Al_2O_3 as the seed to heterogeneously grow the NC precursors into a core-shell structure, which all of the Al_2O_3 source is located in the core and covered by the NC precursors (**Fig. 1** in maintext). Afterwards, a typical solid-state calcination process was used to lithiate the resulting precursors to the final cathode material. To be noted, the calcination temperature we choose is much lower than that used in industrial procedures. Given the fact that the diffusivity of Al atom is considerably larger than that of Ni at high temperature, the solid-state calcination process would promote the Kirkendall effect diffusion of Al atom to the Ni-rich layered oxide cathode, consequently resulting in a single Kirkendall void at the center of secondary particle and an Al-enriched doping interior structure.

Considering the property advantages of the resulting cathode materials, such as unique structure, gradient Al doping, uniformized internal stress distribution, and more

importantly much improved electrochemical performance, we believe that our strategy to synthesize the resulting cathode materials has a bright future for potential industrial production.

7. In Figure S11, please address where the large delta V is coming from even in early cycling, *c*-LiNi_{0.96}Al_{0.04}O₂ shows larger delta V than *hk*-LiNi_{0.96}Al_{0.04}O₂.

Response: Thank you for your valuable comments. As we discussed in the maintext and the response letter, the resulting *hk*-LiNi_{0.96}Al_{0.04}O₂ sample shows much better mechanical properties brought by the unique structural design than the control sample. Cracks will be easily generated in the *c*-LiNi_{0.96}Al_{0.04}O₂ control sample even in the early cycles, thus leading to side reaction at the newly exposed surface and sluggish charge transport kinetics. Consequently, the polarization behavior in the control sample is more severe than the resulting *hk*-LiNi_{0.96}Al_{0.04}O₂ sample in the early cycling, leading to the large delta V in early cycling. The polarization in the control sample would exaggerate as the cycle numbers prolong, which accounts for the larger delta V as shown **Fig. S11**.

REVIEWERS' COMMENTS

Reviewer #1 (Remarks to the Author):

Thank you for addressing all the comments thoroughly. The authors have given a clear explanation and understanding in the revised manuscript and there is no further recommendation for the improvement.

Reviewer #2 (Remarks to the Author):

The authors have addressed all of my concerns and questions. I recommend the publication of this work in Nature Communications.

Reviewer #3 (Remarks to the Author):

The revised manuscript can meet my requirement for being published in Nature Communications.